# EFFICIENT MOLECULAR CONFORMER GENERATION WITH SO(3) AVERAGED FLOW-MATCHING AND REFLOW

## ABSTRACT

Molecular conformer generation is a critical task in computational chemistry and drug discovery. Diverse generative deep learning methods have been proposed and shown to outperform traditional cheminformatics tools. State-of-the-art models leverage neural transport, employing denoising diffusion or flow-matching to generate or refine atomic point clouds from a prior distribution. Still, sampling with existing models requires significant computational expense. In this work, we build upon flow-matching and propose two mechanisms for accelerating training and inference of 3D molecular conformer generation. For fast training, we introduce the SO(3)-*Averaged Flow*, which we show to converge faster and generate better conformer ensembles compared to conditional optimal transport and Kabsch alignment-based optimal transport flow. For fast inference, we further show that reflow methods and distillation of these models enable few-steps or even one-step molecular conformer generation with high quality. Using these two techniques, we demonstrate a model that can match the performance of strong transformer baselines with only a fraction of the number of parameters and generation steps. The training techniques proposed in this work shows the path towards highly efficient molecular conformer generation with flow-based models.

## 1 INTRODUCTION

Molecular conformer generation is the task to predict the ensemble of 3D conformations of molecules given the 2D molecular graphs (Hawkins, 2017). Generating high quality molecular conformers that fit their natural 3D structures is a crucial task for computational chemistry because many physical and chemical properties (Guimarães et al., 2012; Schwab, 2010; Shim & MacKerell Jr, 2011) are determined by the conformers. In the domain of drug discovery, molecular conformer generation is a prerequisite for both structure-based and ligand-based compound virtual screening applications such as molecular docking (Trott & Olson, 2010) and shape similarity search (Rush et al., 2005). For established computational chemistry molecular conformer generation tools, there is a trade-off between generation speed and the quality/diversity of generated conformers (Axelrod & Gomez-Bombarelli, 2022). For example, enhanced molecular dynamics simulation (Grimme, 2019) can generate diverse conformer by sampling the conformation space rather exhaustively, but is slow due to multiple energy function evaluations. RDKit (Landrum, 2016) and some rule-based tools (Hawkins et al., 2010) are faster but may miss many low-energy conformer and the generation quality can deteriorate when molecule size grows. Therefore, deep learning models are being sought as a potential solution to overcome such trade-off and bring fast, diverse, and high-quality molecular conformer generation.

Many earlier works are based on generative models (Simm & Hernández-Lobato, 2019; Zhu et al., 2022; Luo et al., 2021; Shi et al., 2021; Xu et al., 2022) given the stochastic nature of the molecular conformer generation task. There is also regression model such as GeoMol (Ganea et al., 2021) that operates on the substructures of the molecules. However, established cheminformatics tools such as OMEGA (Hawkins et al., 2010) still has better generation quality with faster sampling speed compared with early deep-learning based methods. Torsional diffusion (Jing et al., 2022) is the first diffusion model that achieves better generation quality than cheminformatics model. By restricting the degree-of-freedom on the torsion angles, torsional diffusion can generate diverse conformers with lightweight model and less number of reverse diffusion steps. Molecular conformer field (MCF) (Wang et al., 2024) is a more recent work that does diffusion directly on the Cartesian

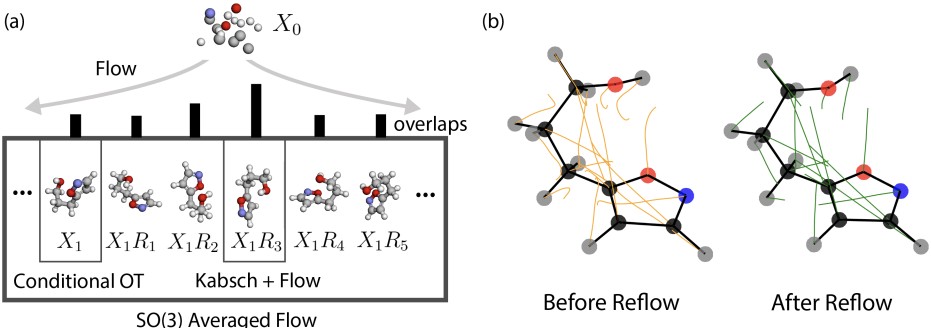

Figure 1: **SO(3)-Averaged Flow and Reflow (a)** We illustrate a comparison between our approach *Averaged Flow*, conditional OT and Kabsch + Flow. While conditional OT randomly assigns any rotation of the data, Kabsch + Flow assigns the rotation of largest overlap. Our method instead computes the expected flow across all rotations. **(b)** Flow trajectory visualization before and after the reflow with 100 Euler steps. The flow trajectories are effectively straightened after reflow.

coordinates of the atoms. With highly scalable transformer architecture, MCF achieves the state-of-the-art conformer generation quality at the cost of tens to hundreds of millions parameters in model size. A more recent work, ET-Flow (Hassan et al., 2024), is also shown to have strong performance by leveraging flow-matching, harmonic prior (Jing et al., 2023), and the Kabsch alignment of the noise and target distribution. With the maturing of diffusion and flow-matching models in the field of molecular conformer generation, the major obstacle that hinders the wide adoption of those models in real-world drug discovery industry is the sampling speed. Iterative ordinary differential equation (ODE) or stochastic differential equation (SDE) solving with large transformer model to generate every conformer can still be computationally infeasible when the library to be virtually screened contains billions of compounds (Bellmann et al., 2022).

In this work, we propose a novel flow-matching training approach to improve the efficiency of deep learning model training and sampling for molecular conformer generation. To improve training efficiency, we design a new flow-matching objective called SO(3)-*Averaged Flow* (Fig. 1a). As an objective, *Averaged Flow* avoids the need to rotationally align prior and data distribution by analytically computing the averaged probability path from the prior to all the rotations of the data sample. Model trained with *Averaged Flow* is experimentally shown to converge faster to better performance. To improve the sampling efficiency, we adopt the *reflow* and distillation technique (Liu et al., 2022) to straighten the flow trajectories (Fig. 1b). Straightened trajectories allow high quality molecular conformer generation with few-step or even one-step ODE solving, thus significantly relieving the computational cost.

Our main contribution can be summarized as: **(i)** Proposed a novel SO(3)-*Averaged Flow* matching objective. *Averaged Flow* eliminates the need of rotational alignment between prior and data by training the model to learn the average probability path over all rotations of the data. *Averaged Flow* leads to faster convergence to better performance for molecular conformer generation, and can be extended to other similar tasks. **(ii)** Introduced reflow with distillation to reduce the number of ODE steps required for the model to generate high quality conformers. Such technique significantly improves the sampling efficiency of flow-matching models in molecular conformer generation.

## 2 BACKGROUND AND RELATED WORK

### 2.1 GENERATIVE MODELS FOR CONFORMER GENERATION

The task of molecular conformer generation in its core is to sample from the intractable conformer distribution conditioned on the 2D molecular graph. Therefore, generative deep learning model is well-suited for such task and many methods have been proposed. Deep learning model are usually trained on datasets containing molecular conformers generated by CREST (Pracht et al., 2020) using computationally expensive semi-empirical quantum chemistry method (Bannwarth et al., 2019) under the hood. Earliest works in this field uses variational autoencoder to generate the intrinsic inter-atomic

distance (Simm & Hernández-Lobato, 2019; Xu et al., 2021). Shi et al. (2021) proposed a score-matching method that learns the gradient of intrinsic atom coordinates in molecular graph. Ganea et al. (2021) started to tackle molecular conformer generation by designing a message passing neural network to predict the local 3D structure and torsion angles. Xu et al. (2022) adopted diffusion model and equivariant graph neural network to generate molecular conformers by iteratively denoising the Euclidean atom coordinates from sampled noise. Torsional diffusion (Jing et al., 2022) reduced the degree-of-freedom by refining the torsion angles of RDKit-generated (Landrum, 2016) initial conformers with a diffusion process on the hypertorus. Such design allowed torsional diffusion to significantly reduce sampling steps. One drawback of torsional diffusion is that it relies on an RDKit-generated conformer as the starting point of diffusion, which adds computational overhead to generation process. The generation quality of RDKit, especially for atom coordinates in rings, can also impact the sample quality of torsional diffusion. Another recent work called DiSCO (Lee et al., 2024a) has proposed to use a Schrödinger bridge-based method to optimize generated conformers. DiSCO can refine molecular conformers generated by any method to lower energy state by aligning the conformational distribution approximated by a prior model to the ground truth distribution. It is shown to improve the conformer generation quality of many methods such as RDKit and even Torsional Diffusion. Molecular conformer field (MCF) proposed by Wang et al. (2024) is a recent work that leverages the scaling power of the transformer architecture (Jaegle et al., 2021) and diffusion model. MCF achieves state-of-the-art performance in molecular conformer generation by training models with tens to hundreds million of parameters to denoise the atoms' Euclidean coordinates using DDPM paradigm (Ho et al., 2020). Equivariant Transformer Flow (ET-Flow) is a concurrent work that trains a equivariant flow-matching model to generate conformers from prior distribution. By combining harmonic prior (Jing et al., 2023), flow-matching, and Kabsch alignment that reduces transport cost, ET-Flow is reported to outperform MCF on several metrics with less ODE steps.

Overall, the trade-off between conformer generation quality and speed is a prevailing issue. Specifically, semi-empirical quantum chemistry can sample very high quality conformers with high computational cost. Diffusion or flow-matching models can generate high quality conformers but the iterative ODE/SDE solving process can be slow, making them less practical for large-scale virtual screening. Cheminformatics tools such as RDKit and OMEGA are very fast but generate conformers with underwhelming diversity.

## 2.2 Flow-matching

*Averaged Flow* is based on Flow Matching (Lipman et al., 2023; Liu et al., 2023a; Albergo & Vanden-Eijnden, 2023), which models a probability density path $p_t(\mathbf{x}_t)$ that gradually transforms an analytically tractable noise distribution ($t = 0$) into a data distribution ($t = 1$), following a time variable $t \in [0, 1]$. Formally, the path $p_t(\mathbf{x}_t)$ corresponds to a *flow* $\psi_t$ that pushes samples from $p_0$ to $p_t$ via $p_t = [\psi]_t * p_0$, where $*$ denotes the push-forward. In practice, the flow is modelled via an ordinary differential equation (ODE) $dx_t = v_t^\theta(x_t)dt$, defined through a learnable vector field $v_t^\theta(x_t)$ with parameters $\theta$. Initialized from noise $x_0 \sim p_0(x_0)$, this ODE simulates the flow and transforms noise into approximate data distribution samples. The probability density path $p_t(x_t)$ and the (intractable) ground-truth vector field $u_t(x_t)$ are related via the continuity equation $dp_t(x)/dt = -\nabla_x \cdot (p_t(x)u_t(x))$. To construct $p_t$ Lipman et al. (2023) introduce a conditional probability $p_t(x|x_1)$ and conditional vector field $u_t(x|x_1)$ both related to their unconditional counterparts as follow:

$$p_t(x) = \int p_t(x|x_1)q(x_1)dx_1. \tag{FM6}$$

$$u_t(x) = \int u_t(x|x_1)\frac{p_t(x|x_1)q(x_1)}{p_t(x)}dx_1 \tag{FM8}$$

With the following simple choices of conditional probability and flow

$$p_t(x|x_1) = \mathcal{N}(x; \mu_t(x_1), \sigma_t^2(x_1)) \tag{FM10}$$

$$\psi_t(x) = \sigma_t(x_1)x + \mu_t(x_1) \tag{FM11}$$

they prove that

$$u_t(x|x_1) = \frac{\sigma_t'(x_1)}{\sigma_t(x_1)}(x - \mu_t(x_1)) + \mu_t'(x_1). \tag{FM15}$$

It is noteworthy that we refer to the linear interpolant $x_t = tx_1 - (1 - t)x_0$ between the noise and data distribution as conditional optimal transport (OT) following Lipman et al. (2023).

## 2.3 RECTIFIED FLOW AND OTHER DISTILLATION

With the success of denoising diffusion probabilistic models (Ho et al., 2020), many attention has been drawn to improve the sampling speed of diffusion models. DDIM (Song et al., 2020) shows that the sampling steps can be significantly reduced by formulating the sampling process as ODE solving. Knowledge distillation techniques (Meng et al., 2023; Salimans & Ho, 2022; Song et al., 2023; Song & Dhariwal, 2023) are also proposed to reduce sampling steps and accelerate generation. Rectified flow (Liu et al., 2022; Liu, 2022) is a method proposed to train the model to learn straight probability flow that bridges prior and data distribution. The *reflow* technique proposed in rectified flow can straighten the flow trajectory and reduce the transport cost, allowing very few-step generation with high quality. After reflow, the model can be further distilled to improve 1-step generation. The reflow and distillation technique has been proven effective in enabling few-step or even single-step text-to-image (Esser et al., 2024; Liu et al., 2023b) and point cloud (Wu et al., 2023) generation.

## 3 METHOD

### 3.1 SO(3)-*Averaged Flow*

The concept of *Averaged Flow* involves recognizing that the data distribution $q$ may exhibit group symmetries, which can be explicitly integrated out. A symmetry group $G$ of $q$ consists of transformations $g : x \mapsto g \cdot x$ that leave the distribution $q$ unchanged, meaning $q(x) = q(g \cdot x)$.

If we focus on Lie groups with a Haar measure, we can express $q$ as

$$q(x) = \int d\hat{x} \, \hat{q}(\hat{x}) \int dg \, \delta_{g \cdot \hat{x}}(x) \tag{1}$$

where $\hat{q}$ represents the distribution over the group orbits, $\hat{x}$ is a representative point of the orbit, and the integral over $G$ uses the Haar measure.

By substituting this into equation equation FM8, we obtain:

$$u_t(x) = \int d\hat{x} \, \hat{q}(\hat{x}) \int dg \, u_t(x|g \cdot \hat{x}) \frac{p_t(x|g \cdot \hat{x})}{p_t(x)} \tag{2}$$

Notice that $p_t(x) = \int d\hat{x} \, \hat{q}(\hat{x}) \int dg \, p_t(x|g \cdot \hat{x})$ is the partition function.

Let's consider the case of conformer generation:

1. $x$ is a $N \times 3$ matrix representing the 3D coordinates of $N$ atoms.

2. The group $G$ is the rotation group $SO(3)$. We will use $R$ to denote the rotation matrix, which acts on $x$ as $x \mapsto xR^T$.

3. The goal is to generate molecular conformers that corresponds to at least local minima in the conformational energy landscape. The orbits $\hat{x}$ in this case corresponds to the different low-energy conformers of a given molecule and their permutations that leave the 2D molecular graph invariant. Therefore, the integral $\int d\hat{x} \, \hat{q}(\hat{x})$ in Eq.2 representing the entire conformer ensemble can be written as $\sum_{\hat{x} \in \text{conformers}} \hat{q}(\hat{x})$, where $\hat{q}(\hat{x})$ is the weight associated to that conformer.

4. $p_t(x|x_1)$ is a Gaussian of the form:

$$p_t(x|x_1) \propto \exp\left( \frac{1}{2} \frac{1}{(1-t)^2} \sum_{ij\delta} (x - tx_1)_{i\delta} \Sigma_{ij} (x - tx_1)_{j\delta} \right) \equiv \exp\left( \frac{1}{2} \frac{\|x - tx_1\|_\Sigma^2}{(1-t)^2} \right)$$

where $\Sigma$ is a $\mathbb{R}^{N \times N}$ matrix.

Let's rewrite $u_t(x)$ in this case:

$$u_t(x) = \frac{1}{Z_t(x,0)} \sum_{\hat{x} \in \text{conformers}} \hat{q}(\hat{x}) \int_{SO(3)} dR \, \frac{\hat{x}R^T - x}{1-t} e^{-\frac{1}{2} \frac{\|x - t\hat{x}R^T\|_\Sigma^2}{(1-t)^2}} \tag{3}$$

where $Z_t(x, \alpha)$ is defined as

$$Z_t(x, \alpha) = \sum_{\hat{x} \in \text{conformers}} \hat{q}(\hat{x}) \int_{SO(3)} dR \, e^{-\frac{1}{2} \frac{\|x - t\hat{x}R^T\|_{\Sigma}^2}{(1-t)^2} + \alpha \cdot (\hat{x}R^T)} \quad (4)$$

with $\alpha$ being an $N \times 3$ matrix that will be needed in the following steps.

Note $u_t(x)$ can be computed by taking the derivative of $\log Z_t(x, \alpha)$ with respect to $\alpha$, and then evaluating it at $\alpha = 0$.

The integral over $R$ can be computed using the formula from Mohlin et al. (2020), which provides a closed-form solution for

$$F \mapsto \log \int_{SO(3)} dR \exp(\text{tr}(FR^T)) \quad (5)$$

where $F$ can be any $3 \times 3$ matrix. In our case, we have

$$\log Z_t(x, \alpha) =$$

$$\log \sum_{\hat{x} \in \text{conformers}} \hat{q}(\hat{x}) \exp \underbrace{\left( \log \int_{SO(3)} dR \exp(\text{tr}(\left(\alpha^T + \frac{t}{(1-t)^2} x^T \Sigma\right) \hat{x} R^T)) + \text{constant in } \alpha \right)}_{\text{closed-form solution using } F = \alpha^T \hat{x} + \frac{t}{(1-t)^2} x^T \Sigma \hat{x}}$$

$$(6)$$

Then we can directly learn

$$\mathcal{L}_{\text{AvgFlow}}(\theta) = \mathbb{E}\left[ \|v_t^\theta(x_t) - u_t(x_t)\|^2 \right], \text{with } t \in [0, 1]. \quad (7)$$

where

$$u_t(x_t) = ([\partial_\alpha \log Z_t(x_t, \alpha)]_{\alpha=0} - x_t)/(1-t) \quad (8)$$

We provide the python implementation of this formula in Appendix A.3.1. Theoretically, $v_t^\theta(x_t)$ can be parameterized by any powerful enough neural network architecture that is capable of learning the conditional OT flow (Lipman et al., 2023).

We note that while our *Averaged Flow* implementation is capable of handling multiple conformer states in the summation in Eq 6. In practice, we approximate the expectation of the conformer ensemble through sampling one conformer in each training epoch. Following previous works (Jing et al., 2022; Wang et al., 2024), the $\hat{q}(\hat{x})$ follows uniform distribution for all conformers. The benchmark of computation time (Table A.3.2) shows that only a small overhead is added when using the *Averaged Flow* objective.

## 3.2 REFLOW AND DISTILLATION

Flow-matching and diffusion-based molecular conformer generation model typically requires hundreds or even thousands steps numerical solving of ODE or SDE during the sampling process. Such iterative process adds computational overhead and hinders the adoption of those model in industrial-level downstream applications, which desire fast generation. One effective technique to reduce the sampling steps without significantly sacrificing the generation quality is to straighten the trajectory. Inspired by the success of such technique in point-cloud generation (Wu et al., 2023) and text-image generation (Esser et al., 2024; Liu et al., 2023b), we finetune our model $v_t^\theta$ trained with Averaged Flow using the *reflow* algorithm proposed in previous rectified flow works (Liu et al., 2022; Liu, 2022). Specifically, we first randomly sample atom coordinates $X_0'$ from standard Gaussian and generates the corresponding conformer $X_1'$ using the Tsitouras' 5/4 solver (Tsitouras, 2011). The coupling $(X_0', X_1')$ is then used in the rectified flow objective to finetune the model:

$$\mathcal{L}_{\text{Reflow}}(\theta) = \mathbb{E}\left[ \|v_t^\theta(X_t', t) - (X_1' - X_0')\|^2 \right], \text{with } t \in [0, 1] \quad (9)$$

Liu et al. (2022) proved that the coupling $(X_0', X_1')$ yields equal or lower transport cost than $(X_0, X_1)$ where $X_0$ is sampled from noise distribution and $X_1$ from data distribution. Therefore, applying the

reflow algorithm to fine-tune model with Eq. 9 can effectively reduce the transport cost and straighten the trajectory.

We empirically find that the transport trajectories bridging Gaussian noise and molecular conformers demonstrates high curvature when $t$ is closer to 0 (Fig. 1b). Therefore, inspired by Lee et al. (2024b), we sample $t$ from a exponential distribution with the probability density function as:

$$p(t) \propto \mathrm{Exp}(\lambda t) \tag{10}$$

where $\lambda$ is -1.2 by selection to focus the training more on $t < 0.5$. The distribution of $t$ is visualized in Fig. 4.

After reflow, the sampling speed can be further reduced by distilling the relation of the coupling $(X_0', X_1')$ into model $v_\theta$ to enable 1-step transport and eliminate the need of ODE solving. During the distillation stage, we fine-tune the reflowed model $v_\theta$ with the following loss function:

$$\mathcal{L}_{\mathrm{Distill}}(\theta) = \mathbb{E}\Big[\|v_t^\theta(X_0', 0) - (X_1' - X_0')\|^2\Big] \tag{11}$$

which is equivalent to the Eq. 9 with $t = 0$.

### 3.3 FLOW-MATCHING MODEL ARCHITECTURE

We use SE(3)-equivariant networks for predicting the time-dependent vector field (Eq. 7). We condition the model on the molecular graph. For implementing our network, we use NequIP model based on Batzner et al. (2022). The features of each atoms and bonds (see Sec. A.1.4 for detailed list of features) are firstly embedded by the model into scalar features. Those features are then mixed with the edge vector through 6 interaction blocks of the model. Lastly a linear layer is used to make prediction of the vector field as type $l = 1$ geometric features. Some noteworthy modifications we made to the original architecture include incorporating edge features to the graph convolution layer and adding residue connection and equivariant layer normalization to stabilize training. Details of our model are provided in Sec. A.1.2 and Fig.5. Overall, the model is trained and fine-tuned using *Averaged Flow* + reflow + distillation following the Algorithm 1. Details of model sampling are included in Sec. A.1.5.

---

**Algorithm 1** *Averaged Flow* with Reflow+Distillation Training

---

**Require:** Molecule Dataset $\mathcal{G} = [G_0, ..., G_D]$, each with conformers $\mathcal{X}^G = [X^{G,0}, ...X^{G,N}]$
**Require:** Learnable Velocity Field Network $v^\theta$
  **1. Base SO(3) Averaged Flow Training**
  $t, X_0, G \sim \mathcal{U}(0, 1), \mathcal{N}(0, 1), \mathcal{G}$
  $X_1 \sim \mathcal{X}^G$
  $X_t \leftarrow t \cdot X_0 + (1 - t) \cdot X_1$
  $u_t(X_t) \leftarrow$ Solve closed-form Eq. 8 for $X_t$ and $t$
  Gradient Step -$\|v_t^\theta(X_t|G) - u_t(X_t)\|^2$
  **2. Reflow**
  $X_0' \sim \mathcal{N}(0, 1)$
  $X_1' \sim \mathrm{ODESolve}\big(v_t^\theta(\cdot|G), X_0'\big)$
  Finetune model with coupled pair $(X_0', X_1')$ through Eq. 9
  **3. Distillation**
  Train model with coupled pair $(X_0', X_1')$ through Eq. 11

---

## 4 EXPERIMENTS

Following previous works, we train and evaluate our model on the GEOM-QM9 and GEOM-Drugs dataset (Axelrod & Gomez-Bombarelli, 2022). We followed the splitting strategy proposed by Ganea et al. (2021); Jing et al. (2022) and test our model on the same test set containing 1000 molecules for both QM9 and Drugs dataset. Dataset and splitting details are included in Sec. A.1.3. The major model evaluation metrics are the average minimum RMSD (AMR, the lower the better) and coverage (COV, the higher the better). Both AMR and coverage are reported for precision (AMR-P and COV-P) and recall (AMR-R and COV-R). The definition of metrics are specified in Sec.A.2.1. Intuitively, coverage

measures the percentage of ground truth conformers being generated (recall) or the percentage of generated conformers being close enough to ground truth (precision), while AMR measures the average RMSD between each ground truth and its closest generated conformer (recall) or vice versa (precision). There are three types of baselines in this work, including *(a)* methods with fast inference speed such as cheminformatics tools (RDKit, OMEGA) and regression model GeoMol(Ganea et al., 2021), *(b)* lightweight diffusion model with reduced degree of freedom (Torsional Diffusion), and *(c)* large transformer-based diffusion or flow model operating on Euclidean atomistic coordinates (MCF and ET-Flow). Moreover, to fairly validate the effectiveness of the *Averaged Flow* objective, we compare the performance of our NequIP-based architecture (Appendix A.1.2) trained with different objectives. Similarly, we compare the performance of the same model architecture before and after reflow+distillation to show the necessity of reflow for few-step generation.

### 4.1 AVERAGED FLOW LEADS TO FASTER CONVERGENCE TO BETTER PERFORMANCE

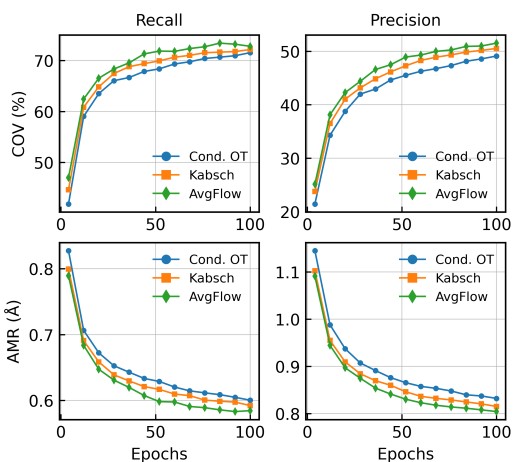

Figure 2: **Model trained with *Averaged Flow* consistently converge to better performance on GEOM-Drugs.** The two objective we compared *Averaged Flow* to are: *(i)* Conditional OT and *(ii)* Kabsch alignment of noise $X_0$ with conformer $X_1$ before conditional OT. Values are the average of a 300-molecule test subset.

To showcase the advantage of the *Averaged Flow* over other training objectives, we evaluate the performance of model trained on different objectives using a randomly sampled GEOM-Drugs test subset containing 300 molecules. The two other objectives to be compared are conditional OT and Kabsch alignment. The Kabsch alignment objective is to rotationally align the sampled noise $X_0$ with conformer $X_1$ before training with the conditional OT objective. Model is evaluated every 8 epochs of training starting from 4 to 100 epochs. Fig. 2 demonstrates that model trained with *Averaged Flow* is consistently better than with both conditional OT and Kabsch alignment on all four metrics. With only 68 epochs of training, *Averaged Flow* has COV-R higher and AMR-R lower than the other two objectives trained for 100 epochs. The COV-P (49.3%) and AMR-P (0.831) of *Averaged Flow* trained for 52 epochs are better than conditional OT (COV-P= 49.1% and AMR-P= 0.832Å) trained for 100 epochs. Also, *Aver-*

*aged Flow* outperforms Kabsch trained for 100 epochs on AMR-P (*Averaged Flow* = 0.814Å and Kabsch= 0.815Å) and on COV-P (*Averaged Flow* = 50.9% and Kabsch= 50.5%) after 76 and 84 epochs, respectively. Overall, model trained with *Averaged Flow* converges with less epochs to better performance in molecular conformer generation.

### 4.2 GEOM-QM9

On the GEOM-QM9 dataset, we compared our model with two prevailingly used cheminformatics tools: RDKit and OMEGA[1], along with GeoMol (Ganea et al., 2021), Torsional Diffusion (Jing et al., 2022), ET-Flow-SS (Hassan et al., 2024), and MCF (Wang et al., 2024). We denote our model trained with *Averaged Flow* as AvgFlow, the model finetuned with reflow as AvgFlow$_{\text{Reflow}}$, and the model further finetuned with distillation as AvgFlow$_{\text{Distill}}$. The number of sampling steps required by diffusion and flow-matching model are also noted. Table. 1 shows that AvgFlow outperforms all other models in the COV-R metrics and almost matching the AMR-R of ET-Flow-SS, indicating it is capable of generating very diverse conformers on the GEOM-QM9 dataset. More importantly, the AvgFlow$_{\text{Reflow}}$ and AvgFlow$_{\text{Distill}}$ achieve higher COV-R than other models with only 2-step and 1-step ODE sampling, respectively. AvgFlow$_{\text{Reflow}}$ also outperforms all cheminformatics tools and GeoMol in all metrics. The benchmark on GEOM-QM9 shows that our model can match the performance of state-of-the-art models with only much less trainable parameters on smaller scale molecule. Table. 1 also shows that reflow+distillation can effectively maintain the conformer generation quality with only 1 or 2 steps of ODE solving.

---

[1]Results adopted from Jing et al. (2022)

Table 1: Quality of ML generated conformer ensembles for GEOM-QM9 ($\delta = 0.5$Å) test set in terms of Coverage (COV) and Average Minimum RMSD (AMR). Bolded results are the best. Baseline values are taken from the corresponding papers. *Due to the use of adaptive step size, the number of steps of AvgFlow is an average value over all test set molecules.

| Method | Step | Recall | | | | Precision | | | |
| --- | --- | --- | --- | --- | --- | --- | --- | --- | --- |
| | | COV (%) ↑ | | AMR (Å) ↓ | | COV (%) ↑ | | AMR (Å) ↓ | |
| | | Mean | Med | Mean | Med | Mean | Med | Mean | Med |
| RDKit | - | 85.1 | 100 | 0.235 | 0.199 | 86.8 | 100 | 0.232 | 0.205 |
| OMEGA | - | 85.5 | 100 | 0.177 | 0.126 | 82.9 | 100 | 0.224 | 0.186 |
| GeoMol | - | 91.5 | 100 | 0.225 | 0.193 | 87.6 | 100 | 0.27 | 0.241 |
| Tor. Diff. | 20 | 92.8 | 100 | 0.178 | 0.147 | 92.7 | 100 | 0.221 | 0.195 |
| ET-Flow-SS (8.3M) | 50 | 95.0 | 100 | **0.083** | **0.035** | 91.0 | 100 | **0.116** | **0.047** |
| MCF-B (64M) | 1000 | 95.0 | 100 | 0.103 | 0.044 | **93.7** | 100 | 0.119 | 0.055 |
| AvgFlow (4.7M) | 60* | **96.4** | 100 | 0.089 | 0.042 | 92.8 | 100 | 0.132 | 0.084 |
| AvgFlow$_{\text{Reflow}}$ (4.7M) | 2 | 95.9 | 100 | 0.151 | 0.104 | 87.7 | 100 | 0.236 | 0.207 |
| AvgFlow$_{\text{Distill}}$ (4.7M) | 1 | 95.1 | 100 | 0.220 | 0.195 | 84.8 | 100 | 0.304 | 0.283 |

## 4.3 GEOM-DRUGS

We then trained and benchmarked our model on GEOM-Drugs, which is a larger dataset containing conformers of drug-like molecules. Table. 2 shows that AvgFlow has good performance on GEOM-Drugs by outperforming torsional diffusion on all metrics. Compared with MCF-S which has approximately 3 times more parameters, our model achieves better COV-P and AMR-P, indicating more AvgFlow-generated conformers are close to ground truth conformers. AvgFlow$_{\text{Reflow}}$ can outperform cheminformatics tools and GeoMol on all metrics, with large margin specifically on the recall metrics. With only 4.7M parameters and 2 ODE steps, AvgFlow$_{\text{Reflow}}$ pushes the limit of the quality-speed trade-off of molecular conformer generations and bears the potential to be adopted for large-scale virtual screen use cases. The AvgFlow$_{\text{Distill}}$ is also shown to achieve better COV-R and AMR-R than cheminformatics tools and GeoMol, showing the our model can maintain high generation diversity even with a single ODE step. The performance of AvgFlow$_{\text{Reflow}}$ drops on precision metrics because of the inevitable model approximation error introduced by the reflow process. More specifically, $X'_1$ generated for reflow may have drifted away from the data distribution and the error is passed on and accumulated during reflow. Therefore, one future direction to improve the performance of reflow and distillation model is to filter the generated $X'_1$ by including only those with low RMSD to ground truth conformers in the reflow finetuning dataset.

Table 2: Quality of generated conformer ensembles for GEOM-DRUGS ($\delta = 0.75$Å) test set in terms of Coverage (COV) and Average Minimum RMSD (AMR). Bolded results are the best. Baseline values are taken from the corresponding papers. *Due to the use of adaptive step size, the number of steps of AvgFlow is an average value over all test set molecules.

| Method | Step | Recall | | | | Precision | | | |
| --- | --- | --- | --- | --- | --- | --- | --- | --- | --- |
| | | COV (%) ↑ | | AMR (Å) ↓ | | COV (%) ↑ | | AMR (Å) ↓ | |
| | | Mean | Med | Mean | Med | Mean | Med | Mean | Med |
| RDKit | - | 38.4 | 28.6 | 1.058 | 1.002 | 40.9 | 30.8 | 0.995 | 0.895 |
| OMEGA | - | 53.4 | 54.6 | 0.841 | 0.762 | 40.5 | 33.3 | 0.946 | 0.854 |
| GeoMol | - | 44.6 | 41.4 | 0.875 | 0.834 | 43.0 | 36.4 | 0.928 | 0.841 |
| Tor. Diff. | 20 | 72.7 | 80.0 | 0.582 | 0.565 | 55.2 | 56.9 | 0.778 | 0.729 |
| ET-Flow-SS (8.3M) | 50 | 79.6 | 84.6 | 0.439 | 0.406 | **75.2** | **81.7** | **0.517** | **0.442** |
| MCF-S (13M) | 1000 | 79.4 | 87.5 | 0.512 | 0.492 | 57.4 | 57.6 | 0.761 | 0.715 |
| MCF-B (64M) | 1000 | 84.0 | 91.5 | 0.427 | 0.402 | 64.0 | 66.2 | 0.667 | 0.605 |
| MCF-L (242M) | 1000 | **84.7** | **92.2** | **0.390** | **0.247** | 66.8 | 71.3 | 0.618 | 0.530 |
| AvgFlow (4.7M) | 102* | 76.8 | 83.6 | 0.523 | 0.511 | 60.6 | 63.5 | 0.706 | 0.670 |
| AvgFlow$_{\text{Reflow}}$ (4.7M) | 2 | 64.2 | 67.7 | 0.663 | 0.661 | 43.1 | 38.9 | 0.871 | 0.853 |
| AvgFlow$_{\text{Distill}}$ (4.7M) | 1 | 55.6 | 56.8 | 0.739 | 0.734 | 36.4 | 30.5 | 0.912 | 0.888 |

## 4.4 WHEN IS REFLOW REALLY NECESSARY?

From the benchmark results on GEOM-Drugs and GEOM-QM9, we understand that our AvgFlow$_{\text{Reflow}}$ model can achieve better performance than cheminformatics methods on all metrics. However, it is obvious that the model's performance drops after reflow especially for the precision metrics. Flow-matching models generally have high generation quality with less steps compared to denoising diffusion model (Lipman et al., 2023) thanks to the ODE sampling process. In this section, we are trying to answer the question: when is reflow really necessary to generate high-quality molecular conformers?

Fig. 3 shows the the performance of our model using Euler sampling method with number of ODE steps $N_{\text{step}} \in \{1, 2, 3, 5, 10, 20, 50, 100\}$. The performance of models are evaluated with the same four metrics on a subset of the GEOM-Drugs test set containing 300 molecules. Overall, AvgFlow has better performance when $N_{\text{step}} \geq 10$ than AvgFlow$_{\text{Reflow}}$. When $N_{\text{step}} < 10$, the performance of AvgFlow has start to collapse and eventually reaches $0\%$ coverage for both recall and precision when $N_{\text{step}} = 1$. The performance gap becomes significant for all metrics when $N_{\text{step}} < 5$. AvgFlow$_{\text{Reflow}}$, on the other hand, has minimal loss in performance until $N_{\text{step}} = 2$ thanks to the straightened flow trajectory. The 1-step generation quality of the model still suffers even after reflow. Distillation can effectively reduce the RMSD of 1-step generated conformers and improve both the COV-R and COV-P. In sum-

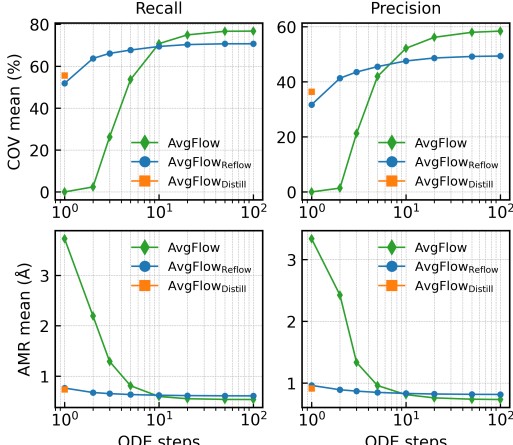

Figure 3: **Effect of the number of ODE steps to model's performance** Comparison between model performance before and after reflow with different number of ODE steps

mary, reflow is critical when generating molecular conformers with very few ODE steps ($N_{\text{step}} < 5$). The reflow and distillation algorithm is model architecture independent, thus can be extended to finetune other powerful models such as ET-Flow (Hassan et al., 2024) to reduce sampling steps.

## 4.5 SAMPLING TIME

Table 3: **Sampling time and performance comparison between models.** Bolded results are the best.

| Method | Step | Time (ms) ↓ | Recall COV (%) ↑ Mean | Recall AMR (Å) ↓ Mean | Precision COV (%) ↑ Mean | Precision AMR (Å) ↓ Mean |
|---|---|---|---|---|---|---|
| Tor. Diff. | 5 | 128 | 58.4 | 0.691 | 36.4 | 0.973 |
| ET-Flow | 5 | 106 | **77.8** | **0.476** | **74.0** | **0.550** |
| MCF-S | 3 | 57.3 | 56.9 | 0.725 | 30.8 | 1.014 |
| MCF-B | 3 | 102 | 66.5 | 0.665 | 39.9 | 0.951 |
| MCF-L | 3 | 134 | 71.6 | 0.636 | 45.3 | 0.686 |
| AvgFlow$_{\text{Reflow}}$ | 2 | **2.68** | 64.2 | 0.663 | 43.1 | 0.871 |

To demonstrate the sampling efficiency of our model, we compared the sampling wall time of our model with MCF and torsional diffusion. Table. 3 shows the sampling time comparison between models[2]. The average sampling time of AvgFlow$_{\text{Reflow}}$ for each conformer in the GEOM-Drugs test set is 2.68 microseconds, which is 21 to $50\times$ faster than different variants of MCF sampled with DDIM for 3 steps. It is also $48\times$ faster than torsional diffusion sampled with 5 steps. AvgFlow$_{\text{Reflow}}$ outperforms MCF-B on precision metrics and reached comparable performance on the recall metrics. AvgFlow$_{\text{Reflow}}$ also outperforms torsional diffusion and MCF-S by large margin with only a fraction of the sampling time. The major speed-up of the our model is due to the JAX implementation and less number of parameters. With reflow ensuring high-quality generation with only 2 ODE steps, our

---

[2]MCF and Torsional Diffusion sampling time values are adopted from Fig.6 of Wang et al. (2024)

model achieves extraordinary sampling efficiency. The 5-steps generation of ET-Flow is achieving better generation quality than all other models. Such high performance is majorly attributed to the harmonic prior (Hassan et al., 2024; Jing et al., 2023). We have further extended the AvgFlow implementation to accommodate the transport from harmonic prior and will explore the effect of harmonic prior in future work. We want to note that the reflow is found to be necessary (Fig. 3) to maintain generation quality when $N_{\text{step}} < 5$, thus making it useful to finetune ET-Flow to improve its sampling speed.

## 5 CONCLUSION

We have presented SO(3)-*Averaged Flow* as a new objective to accelerate the training of flow-matching models for molecular conformer generation. *Averaged Flow* leads to faster convergence to better performance compared with conditional OT and Kabsch alignment. We have also experimented reflow and distillation to straighten the flow trajectory and enable few-step molecular conformer generation. Our model reaches the state-of-the-art performance on the coverage-recall metric of the GEOM-QM9 dataset. It is also matching the performance of transformer-based model which have several times more parameters on the GEOM-Drugs dataset. By analyzing the effect of number of ODE steps to the model generation quality, we find out that reflow and distillation are necessary when very few steps ($N_{\text{step}} < 5$) of conformer generation is desired. Finally, by comparing the sampling time, we demonstrate that our model is approximately 21 to 50 times faster than the other state-of-the-art models, while achieving second to the best generation quality and diversity. Overall, given that the *Averaged Flow* and reflow training scheme can be applied to any models, our method bridges the gap between multi-step flow-matching models and practical molecular conformer generation application by pushing the boundary of quality-speed trade-off.

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

# A APPENDIX

## A.1 EXPERIMENTS DETAILS

### A.1.1 TRAJECTORY AND DISTRIBUTION OF $t$

Here we are visualizing the trajectories of atoms in a molecules during 100-steps of ODE transport. Fig. 1a shows the trajectory before reflow, which demonstrate high curvature at the beginning of the transport ($t$ close to 0). We observed such pattern in trajectory for most of molecules, leading us to sample $t$ from exponential distribution which focus on the $t < 0$ region during the reflow. After reflow, the 100-step ODE trajectory of the same molecules much straighter (Fig. 1b).The distribution of $t$ is visualized in Fig. 4.

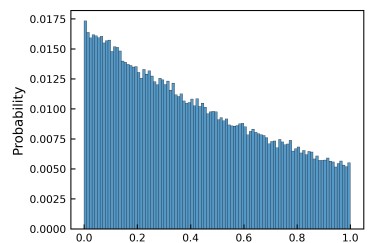

Figure 4: **The distribution of $t$ during reflow**

### A.1.2 MODEL ARCHITECTURE

The equivariant model used in this work is a modified variant (Fig. 5) of the NequIP model (Batzner et al., 2022). The model takes 4 inputs including the atomic features $Z$, relative distance vector between atoms $\vec{r}$, edge (bond) features $e$, and the flow-matching time-step $t$. The output model is a vector field corresponding to the probability flow at $t$. Compared to the original NequIP model, our variant has residue connection and equivariant layer normalization (Liao et al., 2023) after each interaction block, which we found to be highly effective in stabilizing the training of model with more than 4 layers. Bond information in the 2D molecular graph is critical inductive bias for the molecular conformer generation task. To add bond information into the model, we featurize the edges in the molecular graph and concatenate the edge features $e$ with the radial basis embedding of relative distance vector $\vec{r}$. The concatenated message is then fed into the rotationally invariant radial function implemented as an multi-layer perceptron. To keep long-range information in the graph convolution during intermediate time-step $t$, we remove the envelop function from the radial basis and keep only the radial Bessel function.

For both the GEOM-Drugs and GEOM-QM9 dataset, we train a model with 6 interaction blocks. The multiplicity is set to 96 and maximum order of irreps $l$ is 2. The radial function MLP has 2 layers and hidden dimension of 256. Molecular graph are fully-connected with non-bond as an specified bond type. The relative distance vectors are scaled down by a soft cutoff distance of 10Å and 20Å for QM9 and Drugs dataset, respectively. we used 12 Bessel radial basis functions in the model. The model is implemented using `e3nn-jax` (Geiger & Smidt, 2022; Geiger et al., 2022).

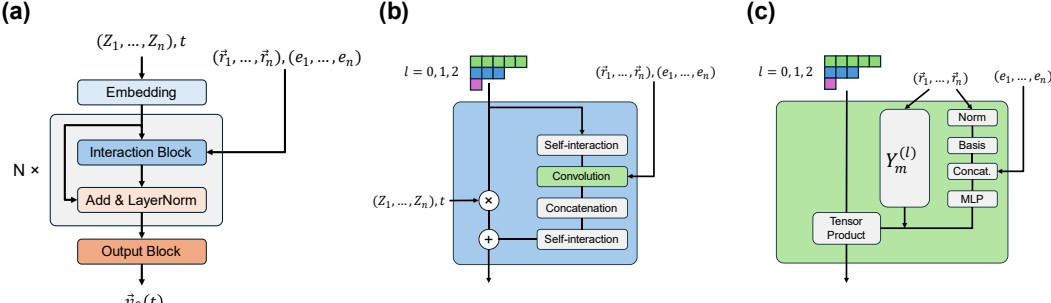

Figure 5: **Model architecture** *(a)* Overview of the modified NequIP architecture for the flow vector field prediction. *(b)* Details of the interaction block, where atomic features are mixed and refined with relative distance vectors $\vec{r}$ and edge features. *(c)* In the convolution block, a learnable radial function MLP incorporate basis embedding of $\vec{r}$ and edge features. Tensor product is used to combine the output of the MLP and the spherical harmonics $Y_m^{(l)}$ projection of $\vec{r}$.

### A.1.3 DATASETS

The dataset we train and benchmark our model on are GEOM-Drugs and GEOM-QM9(Axelrod & Gomez-Bombarelli, 2022). We follow the exact splitting defined and used in previous works (Ganea et al., 2021; Jing et al., 2022; Wang et al., 2024). The train/val/test set of GEOM-Drugs contains 243473/30433/1000 molecules, respectively. The train/val/test set of GEOM-QM9 contains 106586/13323/1000 molecules, respectively.

### A.1.4 MOLECULAR GRAPH FEATURIZATION

We followed the atomic featurization from GeoMol (Ganea et al., 2021). Details of the atomic featurization are included in Table. 4. Graph Laplacian positional encoding vector (Dwivedi et al., 2023) with size of 32 is concatenated with the atomic features for each atom in molecular graph. The edge features is the one-hot encoding of the bond types: {No Bond, Single Bond, Double Bond, Triple Bond, Aromatic Bond}.

Table 4: Atomic features as input to the model

| Name | Description | Range |
|---|---|---|
| atom_type | Atom type | One-hot encoding of the atom type |
| degree | Number of bonded neighbors | $\{x : 0 \leq x \leq 6, x \in \mathbb{Z}\}$ |
| charge | Formal charge of atom | $\{x : -1 \leq x \leq 1, x \in \mathbb{Z}\}$ |
| valence | Implicit valence of atom | $\{x : 0 \leq x \leq 6, x \in \mathbb{Z}\}$ |
| hybridization | Hybridization type | $\{sp, sp^2, sp^3, sp^3d, sp^3d^2, other\}$ |
| chirality | Chirality Tag | {unspecified, tetrahedral CW, tetrahedral CCW, other} |
| num_H | Total number of hydrogens | $\{x : 0 \leq x \leq 8, x \in \mathbb{Z}\}$ |
| aromatic | Whether on aromatic ring | {True, False} |
| num_rings | Number of rings the atom on | $\{x : 0 \leq x \leq 3, x \in \mathbb{Z}\}$ |
| ring_size_3-8 | Whether on ring size of 3-8 | {True, False} |

### A.1.5 TRAINING AND SAMPLING DETAILS

The model is trained with the *Averaged Flow* for 990 epochs on the GEOM-Drugs dataset and 1500 epochs on the GEOM-QM9 dataset using 2 NVIDIA A5880 GPUs. We used dynamic graph batching to maixmize the utilization of GPU memory and reduce JAX compilation time. The effective average batch size is 208 and 416 for Drugs and QM9 dataset, respectively. We used Adam optimizer with learning rate of $1e-2$, which decays to $5e-3$ after 600 epochs and to $1e-3$ after 850 epochs. We selected the top-30 conformers for model training.

To sample coupled $(X_0', X_1')$ for reflow and distillation, we generate 32 noise-sample pairs for each molecule in the Drugs and 64 for each molecule in the QM9 dataset. The reflow and distillation are

done using 4 NVIDIA A100 GPUs and doubling the effective batch size of each dataset. During the reflow stage, the model is finetuned for 870 epochs on Drugs and 1530 epochs on QM9. We used Adam optimizer with learning rate of $5e{-}3$, which decays to $2.5e{-}3$ after 450 epochs for Drugs (500 epochs for QM9), and to $5e{-}4$ after 650 epochs for Drugs (900 epochs for QM9). During the distillation stage, the model is finetuned for 450 epochs on Drugs and 1200 epochs on QM9. We used Adam optimizer with learning rate of $2e{-}3$, which decays to $1e{-}3$ after 300 epochs for Drugs (500 epochs for QM9), and to $2e{-}4$ after 450 epochs for Drugs (900 epochs for QM9). We used exponential moving average (EMA) with a decay of 0.999 for all Averaged Flow, reflow, and distillation training.

To generate the benchmark results of AvgFlow (Table. 1, Table. 1, and Table. 3), we use the Tsitouras' 5/4 solver (Tsitouras, 2011) implemented in the `diffrax` package with adaptive stepping. The relative tolerance and absolute tolerance are set to $1e{-}5$ and $1e{-}6$ when sampling for GEOM-Drugs, respectively. The relative tolerance and absolute tolerance are both set to $1e{-}5$ when sampling for GEOM-QM9. Euler solver is always used for $\text{AvgFlow}_{\text{Reflow}}$ and $\text{AvgFlow}_{\text{Distill}}$. When comparing the effect of ODE steps to models, Euler solver is used.

## A.2 EVALUATION DETAILS

### A.2.1 EVALUATION MTRICS

We report the average minimum RMSD (AMR) between ground truth and generated structures, and Coverage for Recall and Precision. Coverage is defined as the percentage of conformers with a minimum error under a specified AMR threshold. Recall matches each ground truth structure to its closest generated structure, and Precision measures the overall spatial accuracy of the each generated structure. Following Ganea et al. (2021); Jing et al. (2022), we generate two times the number of ground truth structures for each molecule. More formally, for $K = 2L$, let $\{C_l^*\}_{l\in[1,L]}$ and $\{C_k\}_{k\in[1,K]}$ respectively be the sets of ground truth and generated structures:

$$
\begin{aligned}
\text{COV-Precision} &:= \frac{1}{K} \left| \{k \in [1..K] : \min_{l\in[1..L]} \text{RMSD}(C_k, C_l^*) < \delta\} \right|, \\
\text{AMR-Precision} &:= \frac{1}{K} \sum_{k\in[1..K]} \min_{l\in[1..L]} \text{RMSD}(C_k, C_l^*),
\end{aligned}
\tag{12}
$$

where $\delta$ is the coverage threshold. $\delta$ is set to $0.75$Å for the Drugs and $0.5$Å for the QM9 dataset. The recall metrics are obtained by swapping ground truth ($K$) and generated conformers ($L$) in the above equations.

## A.3 AVERAGED FLOW DETAILS

### A.3.1 PYTHON IMPLEMENTATION

Listing 1: Averaged Flow

```python
def avg_harmonic_flow(
    t: jax.Array,  # []
    x: jax.Array,  # [num_nodes, 3]
    x1: jax.Array,  # [num_conformers, num_nodes, 3]
    edges: jax.Array,  # [2, num_edges]
    weights: jax.Array | None = None,  # [num_conformers]
    sigma0: jax.Array = 1.0,
    sigma1: jax.Array = 0.0,
) -> jax.Array:
    degree = jnp.bincount(edges[0], length=x.shape[0])

    def metric(x, y):
        # x and y have shape [num_nodes]
        sigma_t = (1 - t) * sigma0 + t * sigma1
        beta = t / sigma_t**2
        laplacian = jnp.sum(degree * x * y) - jnp.sum(x[edges[0]] * y[edges[1]])
        return beta * laplacian

    avg_x1 = avg_target(x, x1, metric, weights)

    return (avg_x1 - x) / (1 - t)

def avg_flow(
```

```python
        t: jax.Array,  # []
        x: jax.Array,  # [num_nodes, 3]
        x1: jax.Array,  # [num_conformers, num_nodes, 3]
        weights: jax.Array | None = None,  # [num_conformers]
        sigma0: jax.Array = 1.0,
        sigma1: jax.Array = 0.0,
    ) -> jax.Array:
        def metric(x, y):
            # x and y have shape [num_nodes]
            sigma_t = (1 - t) * sigma0 + t * sigma1
            beta = t / sigma_t**2
            return beta * jnp.dot(x, y)

        avg_x1 = avg_target(x, x1, metric, weights)

        return (avg_x1 - x) / (1 - t)

    def avg_target(
        y: jax.Array,  # [num_nodes, 3]
        targets: jax.Array,  # [num_conformers, num_nodes, 3]
        metric: Callable[[jax.Array, jax.Array], jax.Array],
        weights: jax.Array | None = None,  # [num_conformers]
    ) -> jax.Array:
        num_conformers, num_nodes, _ = targets.shape
        assert y.shape == (num_nodes, 3)
        assert targets.shape == (num_conformers, num_nodes, 3)

        def logZ(alpha):
            def f(x):
                # x and y have shape [num_nodes, 3]
                mapped_metric = jax.vmap(jax.vmap(metric, (None, -1)), (-1, None))
                similarity = mapped_metric(x, y)  # [3, 3]
                return logcF(similarity + x.T @ alpha)

            return logsumexp(jax.vmap(f)(targets), weights)

        return jax.grad(logZ)(jnp.zeros_like(y))

    def logsumexp(a: jax.Array, weights: jax.Array | None = None) -> jax.Array:
        assert a.ndim == 1
        assert weights is None or weights.shape == a.shape
        where = (weights > 0) if weights is not None else None

        amax = jnp.max(a, where=where, initial=-jnp.inf)
        amax = jax.lax.stop_gradient(
            jax.lax.select(jnp.isfinite(amax), amax, jax.lax.full_like(amax, 0))
        )
        if where is not None:
            a = jnp.where(where, a, amax)
        exp_a = jax.lax.exp(jax.lax.sub(a, amax))
        if weights is not None:
            exp_a = exp_a * weights
        sumexp = exp_a.sum(where=where)
        return jax.lax.add(jax.lax.log(sumexp), amax)

    # All the code below is adapted from a PyTorch code from David Mohlin, Gerald Bianchi and Josephine Sullivan

    def logcF(F: jax.Array) -> jax.Array:
        # \log \int_{SO(3)} \exp(\text{tr}(F^T R)) dR
        assert F.shape == (3, 3)
        return logcf(*signed_svdvals(F))

    def signed_svdvals(F: jax.Array) -> jax.Array:
        u, s, vh = jnp.linalg.svd(F, full_matrices=False)
        u, vh = jax.lax.stop_gradient((u, vh))
        sign = jnp.sign(jnp.linalg.det(u @ vh))
        return s.at[-1].mul(sign)

    @jax.custom_vjp
    def logcf(s1: jax.Array, s2: jax.Array, s3: jax.Array) -> jax.Array:
        # assume s1 >= s2 >= s3
        s1, s2, s3 = jnp.asarray(s1), jnp.asarray(s2), jnp.asarray(s3)
        return s1 + s2 + s3 + jnp.log(factor(False, s1, s2, s3))

    def _logcf_fwd(
        s1: jax.Array, s2: jax.Array, s3: jax.Array
    ) -> tuple[jax.Array, tuple[jax.Array, jax.Array]]:
        # s1 >= s2 >= s3
        f = factor(False, s1, s2, s3)
        return s1 + s2 + s3 + jnp.log(f), (s1, s2, s3, f)

    def _logcf_bwd(res: tuple[jax.Array, ...], grad: jax.Array) -> tuple[jax.Array]:
        s1, s2, s3, f = res
        # s1 >= s2 >= s3
        assert s1.shape == ()
```

```python
        assert f.shape == ()
        assert grad.shape == ()
        g1 = grad * factor(True, s1, s2, s3) / f
        g2 = grad * factor(True, s2, s1, s3) / f
        g3 = grad * factor(True, s3, s1, s2) / f
        return g1, g2, g3

logcf.defvjp(_logcf_fwd, _logcf_bwd)

def factor(add_x: bool, s1: jax.Array, s2: jax.Array, s3: jax.Array) -> jax.Array:
    def f(x):
        i0 = (1.0 - 2 * x) if add_x else 1.0
        i1 = bessel0((s2 - s3) * x)
        i2 = bessel0((s2 + s3) * (1 - x))
        return i0 * i1 * i2

    tiny = jnp.finfo(s1.dtype).tiny
    a = 2 * (s3 + s1)

    # a non zero:
    a_ = jnp.maximum(a, 0.5)
    y = jnp.linspace(tiny + jnp.exp(-a_), 1.0, 512)
    r1 = jnp.trapezoid(jax.vmap(f)(-jnp.log(y) / a_), y) / a_

    # a (close to) zero:
    x = jnp.linspace(0.0, 1.0, 512, dtype=s1.dtype)
    r2 = jnp.trapezoid(jax.vmap(f)(x) * jnp.exp(-a * x), x)

    return jnp.where(a > 1.0, r1, r2)

def bessel0(x: jax.Array) -> jax.Array:
    p = [1.0, 3.5156229, 3.0899424, 1.2067492, 0.2659732, 0.360768e-1, 0.45813e-2]
    bessel0_a = jnp.array(p[::-1], dtype=x.dtype)

    p = [0.39894228, 0.1328592e-1, 0.225319e-2, -0.157565e-2, 0.916281e-2]
    p += [-0.2057706e-1, 0.2635537e-1, -0.1647633e-1, 0.392377e-2]
    bessel0_b = jnp.array(p[::-1], dtype=x.dtype)

    abs_x = jnp.abs(x)
    x_lim = 3.75

    def w(x, y):
        return jnp.where(abs_x <= x_lim, x, y)

    abs_x_ = w(x_lim, abs_x)

    return w(
        jnp.polyval(bessel0_a, w(abs_x / x_lim, 1.0) ** 2) * jnp.exp(-abs_x),
        jnp.polyval(bessel0_b, w(1.0, x_lim / abs_x_)) / jnp.sqrt(abs_x_),
    )
```

### A.3.2 SPEED BENCHMARK

We benchmarked the time used by our Python implementation to solve the *Averaged Flow* objective for batched graphs. Each graph is set to have 50 nodes (the average number of atoms in GEOM-Drugs molecules is 44). The benchmark is done on a single NVIDIA A5880 GPU.

Table 5: Computation time of *Averaged Flow* on batched graphs (50 nodes per graph). Unit is in ms. $N_{\text{batch}}$ is the number of graphs in a batch and $N_{\text{conformer}}$ is number of conformers used in *Averaged Flow* solving.

| $N_{\text{batch}}$ \ $N_{\text{conformer}}$ | 1 | 10 | 100 | 1000 |
|---|---|---|---|---|
| 1 | 0.6 | 0.5 | 0.5 | 0.6 |
| 10 | 0.5 | 0.5 | 0.6 | 1.0 |
| 100 | 0.5 | 0.6 | 1.1 | 7.6 |
| 1000 | 0.5 | 0.9 | 7.5 | 73.5 |

