# OpenReview forum: "Efficient molecular conformer generation with SO(3) averaged flow-matching and reflow"
_ICLR.cc/2025/Conference — Submitted to ICLR 2025_

### Official Review · Reviewer_4ywq · 2024-10-28

**Soundness:** 2
**Presentation:** 2
**Contribution:** 2
**Rating:** 6
**Confidence:** 3

**Summary:**

This paper proposed SO(3)-Averaged Flow (AvgFlow), which developed the average flow matching objective for faster convergence in training, and applied the technique of reflow and distillation for reduced ODE steps in sampling. AvgFlow achieves comparable performance to the flow matching counterpart with geometrically sophisticated designs in training on QM9, and slightly inferior on GEOM-Drugs.

**Strengths:**

- The authors further explored the application of flow matching for molecular conformer generation, which holds the potential of reducing generation time.
- This paper seems to be the first to develop the SO(3)-average flow matching objective.

**Weaknesses:**

Major:
- The method section is not clear enough. For example in Section 3.1, the authors claimed that the Monte-Carlo estimate of $\hat{x}$ risks introducing a bias, but without further explanation. This is related to Eq (7) where it is not clear on which the expectation is taken. I suggest the authors consider explicitly comparing their derived AvgFlow loss with the original flow matching objective, detailing the differences and highlighting where and why the differences arise. As I believe this constitutes the major part of novelty for this paper, further explanation would help clarify the novelty and contribution.
- This paper lacks enough novelty and empirical contribution. Since the theoretical part for AvgFlow objective is not elaborated thus the contribution is not immediately clear, the remaining reflow and distillation seems a direct application of known techniques in flow matching literature here. This would be okay, though, if the performance is strong enough. However, AvgFlow (together with the reflow and distill variants) appears not surpassing the performance of ET-Flow given comparable NFE (50~100), and the performance of fewer steps is not directly comparable, unless the author also uses the same reflow and distillation over ET-Flow. Therefore, I hold the view that the contribution of this paper is at least not adequately presented.

Minor:
- Line 169 should be \citep
- Line 312 "The Kabsch alignment objective is to rotationally aligning ..." -> "to rotationally align ..." Please check if there are more grammar errors.

**Questions:**

- What is the physical plausibility of generated conformers, i.e. not violating the physical constraint for molecules? For example, the PoseBusters [1] passing rate?
- Can the authors provide a more detailed description of baselines?

[1] https://pubs.rsc.org/en/content/articlehtml/2024/sc/d3sc04185a

---

> ### Author Response · Authors · 2024-11-24
> **Response by the authors**
>
> *We would like to thank the reviewer for their comprehensive review and feedback. We are delighted to see that the reviewer highlights the novelty of our Averaged Flow method in conformer generation. Please see below for the responses to the weaknesses and questions raised by the reviewer:*
>
> >Weakness 1. Details of the *Averaged Flow* methods.
>
> To address the concern of the reviewer about the clarity regarding the Averaged Flow objective, we rewrote the Sec. 3.1 with thorough derivation and explanation on how it is applied to the conformer generation task. We have also attached a code snippet in the Appendix to directly show how the *Averaged Flow* is implemented.
>
> Through the revision of Sec. 3.1, we now set a clear definition of $\hat{x}$ as ground truth conformers. We explain how we train the model in practice by sampling conformers to approximate the expectation of flow to conformer ensembles. The potential bias introduced during the sampling process can be alleviated by factoring the whole conformer ensemble in the calculation of partition function $Z$, which we have implemented in the code snippet (Appendix 3.1).
>
> > Weakness 2. Presentation not clear enough for *Averaged Flow* and lack of novelty with direct application of reflow.
>
> To address the concern of the reviewer about the novelty contribution, we have provided more details about the derivation of the *Averaged Flow* objective as well as the python implementation of it. Furthermore, in Figure 3, we demonstrated that to enable very few step sampling ($N_{step}$<5), reflow is necessary to maintain generation quality. Moreover, the reflow technique is model architecture independent. That being said, with our contribution in this work to confirm the effectiveness of reflow in molecular conformer generation tasks, powerful models such as ET-Flow can also be fine-tuned through reflow to enable faster sampling in future works. We now emphasize this at the end of Sec. 4.4.
>
> >Weakness minor.
>
> We thank the reviewer for pointing out some formatting and grammatical issues. We have thoroughly checked the manuscript and fixed the issues.
>
> >Question 1. Physical plausibility of generated conformers.
>
> Following the suggestion of the reviewer, we tested the PoseBusters passing rate for $\mathrm{AvgFlow}$ (102 steps), $\mathrm{AvgFlow_{Reflow}}$ (2-steps), $\mathrm{Tor. Diff_{3steps}}$ , and $\mathrm{Tor. Diff_{5steps}}$. Taking the $\mathrm{max}(64, 2 \times N_{\mathrm{conformer}} )$ generated conformers of all molecules in the GEOM-Drugs dataset, the average passing rates are 93.8%, 67.1%, 63.6%, 68.7% for $\mathrm{AvgFlow}$, $\mathrm{AvgFlow_{Reflow}}$, $\mathrm{Tor. Diff_{3steps}}$, $\mathrm{Tor. Diff_{5steps}}$, respectively. We can see that $\mathrm{AvgFlow}$ achieves a very high PoseBusters passing rate on generated conformers. 2-step generation of $\mathrm{AvgFlow_{Reflow}}$ can match $\mathrm{Tor. Diff_{5steps}}$ in passing rate, which is promising considering the $\mathrm{Tor. Diff}$ starts from ~100% valid RDKit generated conformers while $\mathrm{AvgFlow_{Reflow}}$ starts from Gaussian noise.
>
> >Question 2. More detailed description of baselines.
>
> We have added more detailed descriptions of baseline to the beginning of Sec. 4. We also emphasized that we demonstrate the effectiveness of both *Averaged Flow* and reflow through comparing the performance of the same model architecture trained with different objectives and before/after reflow. The following paragraph can be found on page 7 of revised manuscript.
>
> >>*There are three types of baselines in this work, including (a) methods with fast inference speed such as cheminformatics tools (RDKit, OMEGA) and regression model GeoMol, (b) lightweight diffusion model with reduced degree of freedom (Torsional Diffusion), and (c) large transformer-based diffusion or flow model operating on Euclidean atomistic coordinates (MCF and ET-Flow). Moreover, to fairly validate the effectiveness of the *AvgFlow* objective, we compare the performance of our NequIP-like architecture (Appendix 1.2) trained with different objectives. Similarly, we compare the performance of the same model architecture before and after reflow+distillation to show the necessity of reflow for few-step generation.*

---

> ### Author Response · Authors · 2024-12-02
>
> Dear reviewer 4ywq:
>
> Since the discussion period will end tomorrow (Dec 2nd), we want to check if our response has addressed your questions and concerns regarding our paper. Please let us know if you have any follow-up comment or question regarding our manuscript. Again, thank you for the time spent on reviewing and discussing the manuscript.

---

> > ### Comment · Reviewer_4ywq · 2024-12-02
> > **Thanks for the rebuttal**
> >
> > I appreciate the authors' answer, which has addressed most of my concerns. I will increase my score but lower the confidence, since I am not an expert in this field, and I still feel that the rewritten Section 3.1 has a large room of improvement. I know this may be a bit last-minute, so I'd like to only list some of the major concerns that the authors might want to consider in their revision.
> >
> > 1. The introduction of Eq. (1) could be better prepared. For example, the authors could introduce the *Lie groups with a Haar measure* a little bit, and provide more citations for unfamiliar readers to refer to, e.g. the general definitions and some application scenarios where they've been adopted, so as to inform the readers why it is necessary and what difficulties it might bring.
> > 2. I'm still a bit confused about what is the authors' derivation and novel contribution and what are some established results such as Eq. (5). Perhaps a clearly outlined overview might help distinguish these two, and highlight the authors' contribution in developing SO(3)-Averaged Flow.
> > 3. Current presentation has mixed the general Averaged Flow with its implementation in conformer generation, leaving the impression that the derived method might just apply to certain circumstances. I understand that every method has its own prerequisite so even if it's true this does not necessarily constitute a weakness, but elaborating the constraint that the method requires to satisfy (for example, I guess some Gaussian distribution?) is necessary for understanding and developing it further.

---

> ### Author Response · Authors · 2024-12-03
>
> We appreciate for the response from the reviewer and score raising. To address the concerns from the reviewer:
>
> >1. More background of *Lie groups and Haar measure*.
>
> We thank the reviewer for bringing this to attention. We would add more background of concept *Lie groups* (our propose objective is flow averaged over SO(3), which is a Lie group) and *Haar measure* (used for defining the integral of data distribution over the SO(3) group). Some citations that will be added to the manuscript if it is accepted include:
>
> >>[1] Anthony Zee. *Group theory in a nutshell for physicists*, volume 17. Princeton University Press, 2016.
> >>
> >>[2] Leopoldo Nachbin and Lulu Bechtolsheim. *The Haar Integral*. 1965.
> >>
> >>[3] Gregory S Chirikjian and Alexander B Kyatkin. *Engineering applications of noncommutative harmonic analysis: with emphasis on rotation and motion groups*. CRC press, 2000.
>
> We hope the references above can help to provide more background information of this specific domain to readers.
>
> >2. For *Averaged Flow*, clearly distinguishing our contribution from prior works.
>
> We want to clarify that our contribution on *Averaged Flow* are three-fold:
> 1. Conceptualized and derived the novel SO(3)-*Averaged Flow* objective for flow-matching training.
> 1. Implemented the closed-form solution of *Averaged Flow*. Here, the closed-form solution of integral over $R$ as proposed by Mohlin et al. (Eq.5) is a critical component in our implementation.
> 1. Validated the effectiveness of Averaged Flow in the case of molecular conformer generation through comprehensive experiments.
>
> >3. Circumstances that *Averaged Flow* can be applied on and constraints.
>
> Theoretically, *Averaged Flow* can be extended to other applications when the generated sample is "correct" no matter how it is rotated. Some examples are protein structure generation and point cloud generation. The constraint here is the Gaussian prior as guessed by the reviewer because the implementation is built upon the closed-form solution in Eq.5. Some variants of standard Gaussian prior can also be accepted including the Harmonic Prior (non-isotropic Gaussian based on covalent bond between atoms). We have included the implementation of *Averaged Flow* with Harmonic Prior in the Appendix as well. We will elaborate on application circumstances and constraints in the manuscript if it is accepted.
>
> *Again, we thank the reviewer for engaging in the discussion. We hope that our response can clarify the concerns of the reviewer and improve the confidence of rating.*

---

### Official Review · Reviewer_9avF · 2024-10-31

**Soundness:** 3
**Presentation:** 2
**Contribution:** 3
**Rating:** 8
**Confidence:** 3

**Summary:**

The authors propose a novel flow matching training objective termed AvgFlow that incorporates the rotation symmetry in the loss objective to find a suitable path embodied as expected flow across all possible rotations from a prior distribution to the target distribution. Their experiments suggest that the AvgFlow objective results into faster training convergence and improved molecular geometries compared to recent flow matching objective which are the conditional OT and conditional OT + Kabsch alignment.
Furthermore, the authors show that the inference process can be accelerated by using the Reflow procedure, to straighten the velocity paths during the ODE trajectory, and in extreme cases perform distillation to achieve a 1-step conformer generator.

**Strengths:**

The paper is well motivated to increase efficiency of flow matching models for 3d molecule conformer generation and fairly compared to the current SOTAs.

The experimental section to compare their AvgFlow objective against other flow matching objective *within* the same Nequip architecture is sound and isolates the effect of the training objective. The further analysis on Reflow and Distillation is also interesting and provides valuable insights for generative modeling in the field of molecular/conformer design.

**Weaknesses:**

The paper lacks sufficient detail regarding the loss function for the (conditional) flow matching objective in Eq. (7).
While the authors mention that the target $u_t(x_t) = \frac{\partial}{\partial \alpha} \log Z_t(x, \alpha) |_{\alpha=0}$ I was not able to find the required information how to solve the integral from Equation (6) also when looking into https://arxiv.org/pdf/2006.09740.
It would be helpful to either provide the code as supplementary information or state in the appendix how the regression target can be computed.

**Questions:**

Since I have raised my concerns about the lack of details in the Weaknesses section, it would be helpful if the authors could answer the following questions:

1) How can I understand $\hat{x}$ in Eq. (1) ? It is mentioned that $\hat{x}$ is a representative point of the orbit (Line 186) - Is it simply a ground truth conformer from the dataset? I.e. a molecule in a given orientation.
2) What are the $\{(A_t, b_t)\}$ exactly and what is their dimensionality ? If $x_1 \in \mathbb{R}^{N/times3}$, $A_t$ can either be a scalar or an $N \times N$ matrix? Is $b_t$ a constant $N\times3$ matrix?
3) Equation (4) is not clear with respect to the dimensionality in the potential energy from the exponential function. Which norm are you using ? Is it a matrix norm?
4) How fast/expensive is it to compute the regression target ? In the Kabsch and cond-OT flow work presented by Hassan et al. (2024), usually an SVD has to performed - It seems from the Appendix in https://arxiv.org/pdf/2006.09740 in (34) multiple computational steps are also required to solve the integral.
5) Why wouldn't you simply increase the model size from AvgFlow to ~8M parameter and see how the model performs? This would be also a fairer comparison (on your side) against ET-Flow-SS. While the results in Figure 2 show that AvgFlow show improved performance against Cond. OT and (Cond OT. + Kabsch), the harmonic prior used in ET-Flow-SS might be also beneficial in your use case.
6) How is the sampling speed against ET-Flow-SS ? (Table 3)

---

> ### Author Response · Authors · 2024-11-24
> **Response by Authors**
>
> *We would like to thank the reviewer for the comprehensive review. We appreciate that the reviewer consider our work to be well-motivated, and our comparison between Averaged Flow and other objectives to be sound. We also acknowledge the reviewers asks for more details of the SO(3)-Averaged flow and implementation. Please see below for the responses to the weakness and questions:*
>
> >Weakness: Implementation detail not clear.
>
> For the reference of the reviewer, we have now updated the manuscript with the Python implementation of *Averaged Flow* in Appendix 3.1.
>
> >1. Definition of $\hat{x}$.
>
> Yes, in the specific problem of molecular conformer generation, $\hat{x}$ is a ground truth conformer from the dataset. To avoid confusion, we have added more explanation in Sec. 3.1 including the definition of $\hat{x}$.
>
> >2. Dimentionality of ($A_t$, $b_t$).
>
> To provide more details and avoid confusion, we have changed some notations and rewriten the *Averaged Flow* derivation in Sec. 3.1. Please refer to the modified manuscript for details.
>
> >3. Clarification of norm in Eq.4.
>
> Yes, we can confirm it's matrix norm.
>
> >4. Computation time of *Averaged Flow* objective.
>
> This question is of high importance as we do not want to add too much computational overhead when training with the new objective. We have attached the Python implementation of *Averaged Flow* in Appendix. We benchmarked the computation time of solving for the averaged flow with batched graphs containing $\\{1, 10, 100, 1000 \\}$ graphs (Table 5 in Appendix 3.2). Each graph in a batch contains 50 nodes, higher than the average number of atoms in GEOM-Drugs molecules. Compared to conditional OT, which computes the flow as linear interpolation between data and noise, our method adds only a very small overhead to the computational time: 0.5 ms with batch size 100 and 1 conformer. It also scales well with the number of graphs in a batch. Therefore, we can conclude that the *Averaged Flow* objective will not be a speed bottleneck during training.
>
> >5. Scaling of model and harmonic prior.
>
> We want to emphasize that we demonstrate the effectiveness of both *Averaged Flow* through comparing the performance of the same model architecture trained with different objectives. That being said, we acknowledge that *Averaged Flow* can be used for training other model architectures such as the one used in ET-Flow. We will surely explore the model architecture and model scaling laws in the future work. We also agree with the reviewer that the harmonic prior can be beneficial for the models’ performance. Therefore, we implemented a variant of *Averaged Flow* that is able to start from the harmonic prior (a non-isotropic Gaussian based on covalent bond). Please refer to the code snippet in Appendix 3.1 for details.
>
> >6. Sampling speed agains ET-Flow-SS.
>
> We have now included the inference speed benchmark of ET-Flow (5-steps) in Table 3. We choose ET-Flow instead of ET-Flow-SS because the checkpoint of ET-Flow has not been released so we can only take the available performance results from the paper. ET-Flow is the only one reported for few-step generation. ET-Flow has very high coverage and precision with 5-steps generation, but we are not able to benchmark its performance with even less ODE steps. As we have shown in Fig.3, $N_{step}$<5 can significantly impact the generation quality. Compared to ET-Flow, our $\mathrm{AvgFlow_{Reflow}}$ is ~40x faster in inference. With the confirmation from our work that reflow can further reduce sample steps, we can explore the possibility of improving ET-Flow sampling speed in the future.

---

> ### Comment · Reviewer_9avF · 2024-11-25
>
> Thank you for your detailed response on my questions and concerns.
> I appreciate your effort in updating the manuscript with the details to better understand what is going on under the hood when performing *Averaged Flow* in Section 3.1 as well as the appendix by providing the python code.
>
> I also believe that the harmonic prior is a strong inductive bias from which Averaged Flow can benefit, regardless of the used model architecture. As I mentioned before, I appreciate the authors effort to evaluate the proposed Average Flow training objective within the same network architecture against OT as well as OT+Kabsch Alignment. I believe the scaling the network combined with Harmonic Prior is a valuable avenue for future work.
>
> Since other reviewers mentioned the lack of details of the implementation in the first round of rebuttal phase, and the authors have resolved this issue, I am willing to increase my score to 8.

---

> > ### Author Response · Authors · 2024-11-25
> >
> > We appreciate that the reviewer acknowledge our improvement of the manuscript over details of *Averaged Flow*. We will surely continue to work on model architecture, scaling, and training with Harmonic Prior (which is now implemented with *Averaged Flow*) in the future.

---

### Official Review · Reviewer_uDbV · 2024-11-01

**Soundness:** 3
**Presentation:** 3
**Contribution:** 2
**Rating:** 6
**Confidence:** 3

**Summary:**

This paper proposes a SO(3) averaging flow matching for molecular conformer generation and adopts a reflow strategy to accelerate the sampling. The SO(3) averaging flow eliminates the need to align the source and target sample and improves the training efficiency by learning the expectation of the conditional vector field that points to the rotated version of the training example. The empirical results on the GEOM-QM9 and GEOM-DRUGS show that the method could reach the same level of performance but be more parameter efficient. The paper also investigates the necessity of the reflow strategy and how the number of sampling steps affects the model performance.

**Strengths:**

1) The SO(3)-Averaged Flow is a novel concept in which the learned probability path is SO(3) invariant, which could improve the generalization of the flow matching method and is model-agnostic.
2) The ReFlow strategy is investigated for the molecular conformer generation that 21 to 50 times speedup is achieved with a relatively small cost of the performance.
3) The author shows the learning curve the SO(3)-Averaged Flow converges better and much faster than the conditional / Equivariant OT flow matching.
4) The overall presentation is clear and easy to follow.
5) The author highlights a practical trick: the timestep t sampled from the exponential distribution rather than the uniform distribution.

**Weaknesses:**

1) The main concern I have is that with the SO(3) averaged flow, the method could potentially benefit more from scaling the model size since more model capacity is needed to learn the density to be SO(3) invariant. However, the author only shows the method could reach a comparable performance with less number of parameters. This causes a concern about whether the proposed method could be effective in a larger setting either with more model parameters or larger molecules. The current experimental results do not show how the method performs with more parameters.
2) Part of the contribution is applying the reflow strategy, a commonly used method for accelerated sampling, to molecular conformer generation. The results are quite promising on the GEOM-QM9 dataset (Table 1), but on a larger dataset GEOM-Drugs (Table 2) the performance degrades significantly for both 'AvgFlow-reflow' and 'AvgFlow-Distill'. This first poses the concern about the necessity of applying the reflow to the problem with huge performance degradation, and second goes back to the question if the SO(3) averaging flow could benefit more from using a large model on a larger dataset.
3) I am also a bit concerned about the quality-speed tradeoff the author mentions in the paper. Ideally, both the accuracy and inference speed should be pushed at the same time.
4) Section 3.1 is a bit unclear to me. From algorithms 1, it seems like the training is operated by augmenting the training set with different rotations. Hence, I am not quite sure about the meaning of presenting the idea with the symmetry group in section 3.1. If the author wants to justify the theoretical motivation and advantages, the writing could be improved by explicitly highlighting the theoretical results.

**Questions:**

1) Since the method is model-agnostic, how does the method perform using a larger model (transformer architecture from the MCF) from which it could potentially benefit?
2) What is the necessity for applying reflow to the method if there is substantial performance degradation?
3) What's the meaning behind those derivations from section 3.1? Does the author want to justify the method theoretically?

---

> ### Author Response · Authors · 2024-11-24
> **Response by Authors**
>
> *We would like to thank the reviewer for their comprehensive review. We appreciate that the reviewer acknowledges the advantage of both Averaged Flow and reflow in conformer generation. We are also glad that the reviewer consider our presentation is clear and paper is easy to follow. Please see below for responses to the weaknesses or questions raised by the reviewer:*
>
> >1. Scaling and the "model-agnostic" feature of the SO(3)-*Averaged Flow*
>
> We agree with the reviewer that models with more parameters might yield better results compared to our current 4.7M parameters NequIP-like architecture. One of the major goals of our work is to demonstrate that *Averaged Flow* as an alternative training objective can help the model to converge faster to better performance. In that sense, we did not compare across different models, but rather compared the same model trained with different objectives (Fig. 2). Since *Averaged Flow* is a new training objective which does not require specific model architecture design, any modern neural network architecture can theoretically be used to parameterize $v^{\theta}_{t} (x_t)$ in Eq.7. We will surely explore various model architectures as well as investigate the scaling law of models in future work.
>
> >2. Necessity of reflow with performance degradation
>
> The reviewer raised a highly valuable question about the performance drop after reflow. In Fig.3 and Sec. 4.4, we compared the performance of our model before and after reflow at different steps. In short, the conclusion is that reflow is necessary to maintain a reasonably good generation quality when $N_{step}<10$. For any model architecture, reducing the number of ODE steps from 10 to 2 means 5x faster inference. Without reflow, such acceleration is not feasible without overly compromising the generation quality. We also want to highlight that AvgFlow_reflow with 2-steps sampling outperforms all cheminformatics tools. It also outperforms or matches Torsional Diffusion (5-steps), MCF-S (3-steps) and MCF-B (3-steps) with >20x faster inference. We believe the points we listed above are strong evidence that reflow is valuable in pushing the boundary of speed-quality trade-off of conformer generation. Moreover, our work serves as a confirmation that the reflow+distillation algorithm is also effective in straightening the flow trajectory when generating molecular conformers with equivariant GNN. With the findings in our work, models like ET-Flow can also be fine-tuned for faster inference.
>
> The coverage metrics used to evaluate the models’ performance are highly sensitive to model errors accumulated during reflow. Specifically, we use generated samples ($X_1’$) for finetuning the model during reflow. If some generated samples are not within coverage threshold with the ground truth conformers, undesired bias can be introduced during reflow and errors will be accumulated, leading to a performance degradation after reflow. One potential solution we will explore in future works is to filter the generated $X_1’$ by including only those with low RMSD to ground truth conformers in the reflow fine-tuning dataset. We will include the improvement discussion in the manuscript (Sec. 4.3).
>
> >3. Quality-speed trade-off
>
> We agree with the reviewer that an ideal solution would improve both accuracy and inference speed at the same time. However, in practice, the trade-off between generation quality and speed is inevitable. For molecular conformer generation, both cheminformatics/quantum chemistry tools and deep learning methods are facing the same issue. Through reflow, we are demonstrating a method to push the boundaries of such trader-off by significantly reducing the inference speed while maintaining a reasonably good generation quality.
>
> >4. Details of SO(3)-*Averaged Flow*. Is it augmenting the training set with different rotations? More explanation on the theoretical motivation.
>
> We want to clarify that the AvgFlow proposed in our paper is a new flow-matching objective instead of a data augmentation strategy (i.e. adding rotations to ground truth conformers). Our objective is used to train models to learn the averaged flow to all rotations of conformers, which is *analytically* solved. Following the suggestion of the reviewer, we have provided more details in Sec. 3.1 and the code implementation in the Appendix 3.1.

---

> > ### Comment · Reviewer_uDbV · 2024-11-28
> >
> > Thanks for the clarification from the author, especially for Python implementation. I acknowledge that AvgFlow is a novel and interesting idea for solving the OT problem other than equivariant OT.
> >
> > However, it's still unclear how $\frac{d logZ_t(x, \alpha)}{d \alpha} |_{\alpha=0} $ would yield the vector field $u_t(x)$.
> >
> > From the Eq. 4, $\frac{d logZ_t(x, \alpha)}{d \alpha} = \frac{1}{Z_t(x, \alpha)} \sum \hat{q}(\hat{x}) \int_{SO(3)}dR e^{-\frac{1}{2}\frac{||x-t\hat{x}R^T||^2}{(1-t)^2}} e^{\alpha \dot (\hat{x}R^T)} (\hat{x}R^T)$
> >
> > Hence:  $\frac{d logZ_t(x, \alpha)}{d \alpha} |_{\alpha=0} = \frac{1}{Z_t(x, 0)} \sum \hat{q}(\hat{x})  \int _{SO(3)} dR e^{-\frac{1}{2}\frac{||x-t\hat{x}R^T||^2}{(1-t)^2}}  (\hat{x}R^T)$.
> >
> > The result above is not the vector field from the Eq.3
> >
> > Also, how to go from the Eq.4 to Eq.6 could be more detailed in math instead of the python implementation. Given the fundamental contribution of this work, I agree the investigation of the scaling law could be left to future work but theoretical part could be further refined. Therefore, I would raise my score with clear mathematical derivations (from Eq.4 to Eq.6) and the confusion of the vector field computation being addressed.

---

> ### Author Response · Authors · 2024-11-29
>
> We appreciate that the reviewer acknowledge the novelty of *Averaged Flow*. We hope the response below can address the questions raised by the reviewer:
>
> >1. Derivation of $u_t(x)$ from $\left[\partial_\alpha \log Z_t(x_t,\alpha)\right]_{\alpha=0}$
>
> Indeed, $\left[\partial_\alpha \log Z_t(x_t,\alpha)\right]_{\alpha=0}$ is not exactly $u_t(x)$, but the non-trivial part of $u_t(x)$. To be more specific, the $-x$ and $\frac{1}{1-t}$ in the term $\frac{\hat{x}R^{T}-x}{1-t}$ of $u_t(x)$ are trivial when computing the average of this term wrt to $R$.
>
> Eq.8 of the manuscript (shown below) provides detail of how to derive the vector field $u_t(x)$ from $\left[\partial_\alpha \log Z_t(x_t,\alpha)\right]_{\alpha=0}$
>
> >>Eq.8: $u_t(x_t) = (\left[\partial_\alpha \log Z_t(x_t,\alpha)\right]_{\alpha=0} - x_t)/(1-t).$
>
>
> >2. Math detail of Eq.4 to Eq.6
>
> Generally speaking, Eq.6 is derived through expanding the quadratics in the exponent in Eq.4. Specifically:
>
> $$
> \begin{aligned}
> -\frac12 \frac{\| x - t \hat x R^T \|^2_\Sigma}{(1-t)^2}  + \alpha \cdot (\hat{x}R^{T}) &= -\frac12 \frac{\| x - t \hat x R^T \|^2_\Sigma}{(1-t)^2} + \mathrm{tr} (\alpha^T \hat x R^T) \\\\
> &= -\frac12 \frac{\mathrm{tr}(x^T \Sigma x) + t^2 \mathrm{tr}(\hat x^T \Sigma \hat x R^T R) - 2 t \  \mathrm{tr}(x^T \Sigma \hat x R^T)}{(1-t)^2} + \mathrm{tr} (\alpha^T \hat x R^T) \\\\
> &= \text{tr}((\alpha^T + \frac{t}{(1-t)^2} x^T \Sigma) \hat{x} R^T) - \underbrace{ \frac{\mathrm{tr}(x^T \Sigma x) + t^2\mathrm{tr}(\hat x^T \Sigma \hat x)}{2(1-t)^2}}_{\text{Denoted as ``constant in }\alpha\text{" in Eq.6}}
> \end{aligned}
> $$
> Please note that note that $R^T R$ is the identity.
>
> Since we are not able to make revision to the manuscript, we will include the details steps from Eq.4 to Eq.6 in the appendix of camera-ready version of the manuscript if the paper is accepted. Thank you for the understanding.

---

> > ### Comment · Reviewer_uDbV · 2024-12-02
> >
> > Thanks for the authors' response.
> >
> > I think with the existence of these trivial terms, it's not fully convincing to say the vector field is the log derivative of the $Z_t(x, \alpha)$ at $\alpha=0$. If I understand correctly, the $\frac{d logZ_t(x_t, t)}{d \alpha} |_{\alpha=0}$ is trying to directly predict the clean data which is averaged over all rotations.
> >
> > I will raise my score because the overall idea is novel and my concerns are addressed. Besides, I hope the author could run the words properly in the final version if accepted.

---

> > > ### Author Response · Authors · 2024-12-03
> > >
> > > We appreciate the reviewer for acknowledging the novelty of *Averaged Flow*. We agree with the reviewer's understanding that the $\left[\partial_\alpha \log Z_t(x_t,\alpha)\right]_{\alpha=0}$ is the clean data averaged over all rotations and it is not exactly equivalent to the vector field $u_t(x)$.
> > >
> > > In the future revision, we will explicit clarify that the objective vector field $u_t(x)$ is *derived* from $\left[\partial_\alpha \log Z_t(x_t,\alpha)\right]_{\alpha=0}$, but not exactly the same.

---

### Official Review · Reviewer_mA9y · 2024-11-02

**Soundness:** 3
**Presentation:** 2
**Contribution:** 3
**Rating:** 5
**Confidence:** 3

**Summary:**

This paper propose accelerate 3D molecular conformer generation by introducing SO(3)-Averaged
Flow for faster training and using reflow and distillation for rapid, high-quality inference.

**Strengths:**

1. The proposed method balances compute cost and performance.
2. The proposed method removes the need for rotational alignment by averaging over data rotations.
3. Faster inference is achieved.

**Weaknesses:**

Please refer to questions below.

**Questions:**

1. In contribution (iii), it is proposed that Averaged Flow and reflow + distillation are model-independent, and whether there is any relevant
experimental support?
2. To compare the time consumption and computational cost of different steps in ET-Flow, please refer to Table 4 in ET-Flow[1]. If ET-Flow
uses fewer steps, how does the performance change? Is there a significant gap?
[1] ET-Flow: Equivariant Flow-Matching for Molecular Conformer Generation. NeurIPS 2024.
3. In section 3.1, please provide further details on the derivation of the relevant formulas.
4. Could you provide a comparison of different methods and different steps for generating conformers?

---

> ### Author Response · Authors · 2024-11-24
> **Response by Authors**
>
> *We would like to thank the reviewer for their feedback. We appreciate that the reviewer highlights our contribution in balancing compute cost and performance, which is the core motivation of this work. Please see below for the responses to the raised questions:*
>
> >1. Experimental support for "model-independency" of the *Averaged Flow* and reflow+distillation method.
>
> We agree with the reviewer that emphasizing on Averaged Flow and reflow+distillation is “model-independent” requires more experimental support with multiple model architectures being trained using our framework. The focus of the manuscript is to demonstrate the effectiveness of Averaged Flow and reflow in improving training and inference of model for faster conformer generation. Therefore, we removed the third point in the main contribution claim. However, we still believe that both methods can theoretically be extended to other model architectures for several reasons:
>
> 1. Averaged Flow is simply an alternative objective to the condition OT (linear interpolant), which does not require extra design of neural network architecture. Model architectures that are powerful enough to be used in flow-matching tasks (such as ET-Flow) can be used in Eq.7 to parameterize $v^{\theta}_{t} (x_t)$.
>
> 1. Reflow+distillation has been applied to domains other than image generation, such as point cloud generation. Our work serves as a confirmation that the reflow+distillation algorithm is also effective in straightening the flow trajectory when generating molecular conformers with equivariant GNN. With the findings in our work, models like ET-Flow can also be finetuned for faster inference.
>
> 1. That being said, experiments on different model architectures to search for the optimal ones are definitely critical and will be pursued in future works.
>
> >2. Performance and computational cost change of ET-Flow with fewer steps.
>
> We thank the reviewer for raising the question about inference speed comparison between ET-Flow and our model. We tested the inference speed of ET-Flow with 5-steps. To be consistent, the inference speed is benchmarked with batchsize=1. The Inference time of ET-Flow is added to Table 3 of the manuscript. We can see that $\mathrm{AvgFlow_{Reflow}}$ is ~40x faster than ET-Flow$\mathrm{_{5step}}$.
>
> ET-Flow maintains high generation quality with Nstep=5. However, our analysis in Fig.3 shows that the generation quality can drop significantly when $N_{step}<5$. Since the checkpoint of ET-Flow has not been released, we are not able to benchmark its generation quality with $<5$ ODE steps. We hope that the reflow and distillation algorithm in our paper can be extended to finetune powerful models like ET-Flow to further reduce sampling steps and accelerate inference.
>
> >3. Details of SO(3)-*Averaged Flow* derivation.
>
> We have added a thorough explanation of how the new objective is calculated with a closed-form solution. The mathematical derivation is also accompanied by code implementation which is added to the Appendix 2.1 of the manuscript.
>
> >4. Comparison of different conformer generation methods and different steps.
>
> It is a great suggestion from the reviewer to compare methods with different steps of sampling. We want to emphasize that one goal of this work is to enable few-step conformer generation through reflow. Therefore, we focus more on the generation quality with a very limited number of steps $N_{step}<5$. In Fig.3, we demonstrate that the benchmark performance of $\mathrm{AvgFlow_{Reflow}}$ surpasses the model by a large margin when $N_{step}<5$. Then, in Table 3, we compare $\mathrm{AvgFlow_{Reflow}}$ with other models sampled with 3-5 steps and demonstrate that reflow effectively alleviated the performance degradation in few-steps generation.

---

> ### Author Response · Authors · 2024-12-02
>
> Dear reviewer mA9y:
>
> Since the discussion period will end tomorrow (Dec 2nd), we want to check if our response has addressed your questions and concerns regarding our paper. Please let us know if you have any follow-up comment or question regarding our manuscript. Again, thank you for the time spent on reviewing and discussing the manuscript.

---

### Official Review · Reviewer_GPaB · 2024-11-03

**Soundness:** 2
**Presentation:** 3
**Contribution:** 2
**Rating:** 5
**Confidence:** 4

**Summary:**

The authors present AvgFlow, a framework for molecular conformer generation based on Flow Matching technology. The proposed approach aims to achieve rotation-awareness and faster sampling through two key improvements: SO(3)-averaging and a reflow strategy. Integrating and averaging over symmetrical groups of target conformers, the model aims to generalize on the SO(3) group. The authors show that the model converges faster compared to conditional OT and Kabsch alignment-based flow matching. Additional application of the Reflow strategy and further distillation allows the model to perform sampling with as few as two and one steps, respectively.

**Strengths:**

The paper introduces promising approach by leveraging Conditional Flow Matching and SE(3)-equivariant networks, along with averaging on SO(3) group and Reflow-distillation strategies that accelerate sampling. The integration of these technologies is cutting-edge, especially in the domain of molecular conformer generation — a critical task in drug discovery. The writing is mostly clear, and the overall framework reflects state-of-the-art approaches applied to an important problem.

**Weaknesses:**

Although the framework integrates several advanced components, each of these techniques is already actively explored in closely related domains, such as proteins and point clouds. There appears to be limited novelty in the adaptation of these methods for the conformer generation task. Moreover, the theoretical rationale and empirical evidence supporting the inclusion of each component could strengthen the novelty of the manuscript. The necessity of using SO(3)-averaging is not fully justified, and the reflow strategy, while reducing sampling steps, results in a noticeable decline in generation performance, without further discussion on the sampling step-generation quality trade-off. These concerns are described further in the questions below.

**Questions:**

Below are some key points requiring further clarification or revision:

1. The paper mentions that the SO(3)-averaged flow is applicable when the integral over all rotations is tractable, though real data often do not meet this condition. However, the implementation details are sparse. How are the rotations sampled, and how many are used? Have the authors performed the hyperparameter search for such numbers? Since SO(3)-averaging is a core component, it is essential to clearly explain the sampling and averaging procedures. Authors should explain their implementation details in reference to FrameAveraging [1], one of the examples of similar application of rotation averaging in point cloud domain.  Additionally, the absence of source code limits transparency and reproducibility. Sharing the code—via supplementary materials or an anonymous repository—would enhance the AI community’s understanding of the framework and align with best practices for open research.

2. The model is built on the NequIP backbone, which is SE(3)-equivariant. Given this, it seems the network should already handle rotations without requiring explicit SO(3)-averaging. This additional step may function primarily as a data augmentation strategy, raising concerns about its necessity. A more thorough explanation is needed to justify why SO(3)-averaging is essential, beyond what the SE(3)-equivariant model offers by design.

3. The paper refers to the model as "model agnostic," but there is no clear discussion on how the framework transforms distributions from arbitrary priors. For example, DiSCO [2] recently proposed a Schrödinger bridge-based approach for conformer generation that achieves this transformation. It would be valuable for the authors to compare their approach with DiSCO, at least conceptually, to clarify how their framework embodies the claimed "model agnostic" property, which should be supported by how sampling can be performed from an arbitrary distribution instead of Gaussian prior.

4. While the reflow strategy and distillation techniques reduce the sampling steps, they also result in a significant drop in performance, especially in the GEOM-DRUGS dataset (Table 2). It is crucial to provide a deeper discussion on this trade-off, particularly since balancing performance with sampling efficiency remains a key challenge. Currently, the paper does not provide sufficient supporting evidences that the framework adequately addresses this balance.

5. In the section titled “AVERAGED FLOW LEADS TO FASTER CONVERGENCE TO BETTER PERFORMANCE,” the authors present epoch-based comparisons to demonstrate the efficiency of SO(3)-averaging. However, if each conformer undergoes multiple rotations before averaging, this effectively multiplies the training data per epoch. This process is similar to training with augmented data, which complicates comparisons with models that use only a single orientation per epoch. A more meaningful comparison would be based on training time or the number of batch iterations, rather than epochs.

References

[1] Puny, Omri, et al. "Frame Averaging for Invariant and Equivariant Network Design." *International Conference on Learning Representations* (2022).

[2] Lee, Danyeong, et al. "Disco: Diffusion Schrödinger bridge for molecular conformer optimization." *Proceedings of the AAAI Conference on Artificial Intelligence*. Vol. 38. No. 12. 2024.

---

> ### Author Response · Authors · 2024-11-24
> **Response by Authors. Part 1**
>
> *We would like to thank the reviewer for the thorough review and acknowledging our technical contribution. Please see below for the point-to-point response to questions raised by the reviewer:*
>
> >1. SO(3)-*Averaged Flow* details, implementation, and comparison to FrameAveraging.
>
> We agree with the reviewer that sharing the code of Averaged Flow implementation can help to clarify the sampling and averaging procedure. We have now attached the code in Appendix 2.1.
>
> We have to clarify that the Averaged Flow objective is analytically calculated without sampling rotations. The rewritten Sec. 3.1 of the manuscript provides more detailed mathematical derivation. We hope the detailed derivation along with the code can improve the transparency of our method to readers.
>
> We thank the reviewer for bringing the FrameAveraging paper to attention. Based on our understanding, the goal of FrameAveraging is to adapt backbone networks to be invariant or equivariant to desired symmetries. FA achieves that by averaging a subset of group operators (called Frame) which they show yield the same equivariance/invariance as averaging over the entire group. In contrast, the goal of *Averaged Flow* is not to change symmetry equivariance/invariance of the backbone model, but rather to variance reduced training objective. Unlike FA that averaged over a subset of group operators, *Averaged Flow* analytically computes the averaging of SO(3) with closed-form solution.
>
> >2. Explanation needed to justify SO(3)-*Averaged Flow* when backbone is equivariant.
>
> We want to clarify that the AvgFlow proposed in our paper is a new flow-matching objective instead of a data augmentation strategy (i.e. adding rotations to ground truth conformers). Our objective is used to train models to learn the averaged flow to all rotations of conformers, which is analytically solved. To clarify the confusion, we provided more details in Sec. 3.1 and code implementation in Appendix 2.1.
>
> For the conformer generation task, the generated sample can be freely rotated while still being a “correct” conformer. Starting from a rotationally-invariant distribution such as standard Gaussian, the vector field (flow) of atoms at time-step t can point to any rotation of the ground truth conformer and still be “correct”. Therefore, instead of data augmentation by randomly rotating the ground truth conformers during training, we analytically solved for the SO(3)-averaged flow and used that as an alternative variance-reduced training objective. The AvgFlow objective helps the model to converge faster to better performance compared with the linear interpolation or Kabsch-aligning objective to the starting noise.
>
> >3. Explanation of *Averaged Flow* being "model-agnostic" and comparison to other methods such as DiCSO.
>
> Since the AvgFlow objective proposed in our manuscript is a training objective, any modern neural network architecture can theoretically be used to parameterize $v^{\theta}_{t} (x_t)$ in Eq.7. To clarify, the phrase “model agnostic” specifically means “architecture-agnostic” instead of claiming AvgFlow can transport atom coordinates sampled from any arbitrary distribution to ground truth conformational distribution.
>
> We thank the reviewer for bringing up the DiSCO paper. We believe that DiSCO can be very useful to optimize conformers generated by a training model or cheminformatics tools to lower-energy states. From our perspective, DiSCO serves more as an optimization tool like xTB relaxation instead of generating from pure Gaussian noise. We have included the discussion of DiSCO in the related work section.

---

> ### Author Response · Authors · 2024-11-24
> **Response by Authors. Part 2**
>
> >4. Necessity of reflow+distillation.
>
> The reviewer raised a highly valuable question about the performance drop after reflow. In Fig.3 and Sec. 4.4, we compared the performance of our model before and after reflow at different steps. In short, the conclusion is that reflow is necessary to maintain a reasonably good generation quality when $N_{step}<10$. For any model architecture, reducing the number of ODE steps from 10 to 2 means 5x faster inference. Without reflow, such acceleration is not feasible without overly compromising the generation quality. We also want to highlight that $\mathrm{AvgFlow_{Reflow}}$ with 2-steps sampling outperforms all cheminformatics tools. It also outperforms or matches Torsional Diffusion (5-steps), MCF-S (3-steps) and MCF-B (3-steps) with >20x faster inference. We believe the points we listed above are strong evidence that reflow is valuable in pushing the boundary of speed-quality trade-off of conformer generation.
>
> The coverage metrics used to evaluate the models’ performance are highly sensitive to model errors accumulated during reflow. Specifically, we use generated samples ($X_1’$) for fine-tuning the model during reflow. If some generated samples are not within coverage threshold with the ground truth conformers, undesired bias can be introduced during reflow and errors will be accumulated, leading to a performance degradation after reflow. One potential solution we will explore in future works is to filter the generated ($X_1’$) by including only those with low RMSD to ground truth conformers in the reflow fine-tuning dataset. We will include the improvement discussion in the manuscript (Sec. 4.3).
>
> >5. Computation time of SO(3)-*Averaged Flow*
>
> We thank the reviewer for the suggestion. To clarify, instead of sampling multiple rotations of training data before averaging, we analytically solve for the SO(3)-*Averaged Flow* (implementation details added to the Appendix) and use that as the training objective. Therefore, the training data size is not increased during each epoch of training.
>
> We benchmarked the computation time of solving for the averaged flow with batched graphs containing $\\{ 1, 10, 100, 1000 \\}$ graphs (Table 5). Each graph in a batch contains 50 nodes, higher than the average number of atoms in GEOM-Drugs molecules. Compared to conditional OT, which computes the flow as linear interpolation between data and noise, our method adds only a very small overhead to the computational time: 0.5 ms with batch size 100 and 1 conformer. It also scales well with the number of graphs in a batch. Therefore, we can conclude that the Averaged Flow objective will not be a speed bottleneck during training.

---

> ### Author Response · Authors · 2024-12-02
>
> Dear reviewer GPaB:
>
> Since the discussion period will end tomorrow (Dec 2nd), we want to check if our response has addressed your questions and concerns regarding our paper. Please let us know if you have any follow-up comment or question regarding our manuscript. Again, thank you for the time spent on reviewing and discussing the manuscript.

---

### Author Response · Authors · 2024-11-25
**Overall Response by Authors**

We would like to thank all reviewers for their comprehensive reviews and constructive feedback. We are delighted to see that the role of reflow in balancing computational cost and generation quality is generally acknowledged. We are also glad to see the novelty of the SO(3)-Averaged Flow objective is highlighted by several reviewers.

*Considering the feedback from reviewers, we made multiple modifications and improvements to the manuscript, including:*

1. We have **rewritten the Sec. 3.1 of the manuscript to provide more details of the derivation of SO(3)-*Averaged Flow* objective**. In the revised section, we emphasized that *Averaged Flow* is a training objective calculated with the closed-form solution of the averaged probability flow over the entire SO(3) group. We also provided more explanation of how the mathematical derivation is related to the specific conformer generation application.

1. We have **attached the Python implementation of *Averaged Flow* to the manuscript**. The computation time of *Averaged Flow* is benchmarked with batched graphs of various sizes to demonstrate the small computation overhead need to use *Averaged Flow* objective. We have also extended *Averaged Flow* to be compatible with Harmonic Prior besides the regular Gaussian distribution prior. Please refer to Appendix 3.1.

1. **More details of baselines have been added**. We have discussed the baseline choices more thoroughly in the manuscript, included more related work in conformer optimization, and added inference time benchmark of ET-Flow.

1. We have **further discussed the potential cause of performance drop after reflow**, and proposed a strategy to potentially mitigate such an issue in the future. Meanwhile, we emphasized the necessity of reflow in maintaining performance with very few (<5) steps conformer generation.

*We believe that these additions further improve our work, and we would like to thank the reviewers for the suggestions.*

---

### Meta-Review · Area_Chair_iiJa · 2024-12-22

**Metareview:**

The paper works on speeding up the training and generation process of molecular conformation using flow matching generative models. In order to improve the training time, it proposes an analytical rotation-averaged flow to improve the training efficiency and it then uses and distills reflow models for faster generation (one or two steps) at test time. The model is tested on two favorably benchmarks against conditional and equivariant optimal transport flows.

The reviewers appreciated the novelty and relevance of the SO3-averaged flows and the use of reflow within the context of conformer generation, the better and quicker generalization of the new training scheme compared to the baselines and the fast (one or two step) generation at test time.

The reviewers also raised concerns regarding the technical novelty compared to prior work which has used averaging and introduced or used reflow and distillation, lack of empirical evidence for the scalability of the method to larger models and large molecules, the apparent degradation of the performance using reflow and its trade-off with the generation quality, the choice of hyperparameters especially when dealing with general problems where the integration over SO3 may not be closed form, comparison to data augmentation both empirically but also conceptually to draw insights into the differences, details of the derivations and motivation of some formal results, proper comparison of convergence time with baselines, and finally, lack of theoretical or empirical evidence for model independence,

The authors provided a thorough rebuttal where they argued that their method is different from prior work that use averaging as a means to achieve invariance to symmetry groups and instead it is used to reduce the variance of the model whereby the generalization and its speed is optimized. They further pointed out that the averaging is done analytically and not through multiple sampling of the flow. They also provided a comparison of the training time in absolute wall-clock time that shows no significant increase for the SO3-integrated average flow. Regarding “model independence” the authors argued that by model, they mean the backbone architecture, and discussed why the proposed method is independent of such a choice but they also removed this point as a contribution since no experimental evidence is provided with different architectures to support this. They promised to rectify the unclarity and imprecision of the formal results in the next version.

Most reviewers participated in the discussion, but not all. For the main undiscussed point, the AC agrees with the authors’ arguments regarding the proposed method being different to frame averaging techniques both in terms of the mechanism and the purpose and finds the empirically negligible added computation time per step of averaged-flow plausible.

Overall, the paper, even after discussions with authors and among reviewers themselves remain at the borderline. After careful consideration of the paper, the reviews, the rebuttal, and the final discussions, the AC recommends rejection despite its merits. The AC believes the paper can be divided into two works, one in improving the training time where a technically novel objective is contributed to the field and the other is to improve the inference time where established approaches of reflow+distillation are adopted which importantly, adversely, affects the training time. Therefore the two aspects do not seem to contribute to one scientific objective. Importantly, as evidenced by the reviewers’ feedback, each of those two aspects needs more work to become a clear conclusive contribution and putting them together does not help either of the messages. While the first part is novel it needs more empirical support and more precision in the derivations, the second part is mainly application of established methods on fast conformer generation which then would need a thorough analysis across different methods, architectures, datasets, and benchmarks. The AC believes the first part/aspect is clearly novel and interesting to the field and can become a viable submission on its own but, as is clear in all reviews, it should be consolidated, for instance, first by improving the presentation and motivation of the formal results and second by more experimental evidence particularly to demonstrate its scalability and applicability to either different architectures or tasks —ideally both.

**Additional Comments On Reviewer Discussion:**

The paper was reviewed by five expert reviewers, mostly remaining at the borderline despite the rebuttal by the authors. The reviewers leaning on the positive side did still raise concerns similar, in the spirit, to other more-negative reviewers. The AC considered all materials thoroughly and sided with the reviewers that leaned towards rejection despite the merits of the work. The reviewers covered a range of directly-relevant expertise from flow matching to conformer generation.

---

### Decision · Program_Chairs · 2025-01-22

Reject